

# Retrieval of Aerosol Fine-mode Fraction over China from Satellite Multiangle Polarized Observations: Validation and Application

Yang Zhang[1], Zhengqiang Li[2], Zhihong Liu[1], Yongqian Wang[1, 3], Lili Qie[2], Yisong Xie[2], Weizhen Hou[2], Lu Leng[4]

[1]College of Resources and Environment, University of Information Technology, Chengdu 610225, China
[2]State Environmental Protection Key Laboratory of Satellite Remote Sensing, Aerospace Information Research Institute, Chinese Academy of Sciences, Beijing 100101, China
[3]Chongqing Institute of Meteorological Sciences, Chongqing 401147, China
[4]Beijing Enterprises (Chengdu Shuangliu) Water Co., Ltd., Chengdu 610000, China

*Correspondence to*: Zhengqiang Li (lizq@radi.ac.cn)

**Abstract.** The aerosol fine-mode fraction (FMF) is an important optical parameter of aerosols, and the FMF is difficult to accurately retrieve by traditional satellite remote sensing methods. In this study, FMF retrieval was carried out based on the multiangle polarization data of Polarization and Anisotropy of Reflectances for Atmospheric Science coupled with Observations from Lidar (PARASOL), which overcame the shortcomings of the FMF retrieval algorithm in our previous research. In this research, FMF retrieval was carried out in China and compared with the AErosol RObotic NETwork (AERONET) ground-based observation results, Moderate Resolution Imaging Spectroradiometer (MODIS) FMF products, and Generalized Retrieval of Aerosol and Surface Properties (GRASP) FMF results. In addition, application of the FMF retrieval algorithm was carried out, a new FMF dataset was produced, and the annual and quarterly average results of FMF from 2006 to 2013 were obtained in all of China. The research results show that the FMF retrieval results of this study are comparable with the AERONET ground-based observation results in China, with correlation coefficient (r), mean absolute error (MAE), root mean square error (RMSE), and the proportion of results that fall with the expected error (Within EE) are 0.770, 0.143, 0.170, and 60.96%, respectively. Compared with the MODIS FMF products, the FMF results of this study are closer to the AERONET ground-based observations. Compared with the FMF results of GRASP, the FMF results of this study are closer to the spatial variation in the ratio of $PM_{2.5}$ to $PM_{10}$ near the ground. The analysis of the annual and seasonal average FMF of China from 2006 to 2013 shows that the FMF high value area in China is mainly maintained in the area east of the "Hu Line", with the highest FMF year being 2013, and the highest FMF season is winter.

## 1 Introduction

Aerosols have a great impact on human production and life and climate change (Kaufman et al., 2002;Huang et al., 2014;Shi et al., 2018). Aerosols have become a research hotspot for scientists from various fields. There are many methods for monitoring aerosols, among which the large-scale coverage of remote sensing technology makes it an effective method for monitoring aerosols. Aerosols produce strong scattering effects in the visible light band (Kokhanovsky et al., 2015). Therefore,





in current satellite remote sensing, visible light channels are generally used to observe aerosols, and aerosol information can be obtained on a global scale. At present, in the field of atmospheric environmental research, aerosol optical depth (AOD) products produced by traditional satellite remote sensing platforms, such as Moderate Resolution Imaging Spectroradiometer

(MODIS), are the most commonly used (Bellouin et al., 2005;Lee et al., 2011;Xie et al., 2015;Zhao et al., 2017;Zhang et al., 2020). Related scholars have carried out many AOD retrieval studies on traditional scalar observation platforms, which can achieve high-precision retrievals and retrieval of AOD (Li et al., 2013;Kim et al., 2014;Zhang et al., 2014;Zhong et al., 2017;Ge et al., 2019). However, other new aerosol optical parameters, such as the fine-mode fraction (FMF), have poor retrieval accuracy over land, which is quite different from the ground-based observation results (Levy et al., 2010). The FMF is a

parameter that can reflect the content of human-made aerosols (Bellouin et al., 2005;Kaufman et al., 2005), and application requirements have been put forward in many studies. For example, in the particle remote sensing (PMRS) model based on the pure physical approach proposed by Zhang and Li, FMF is one of the core input parameters that determines the final particle concentration retrieval accuracy (Zhang and Li, 2015;Li et al., 2016). However, the existing publicly released satellite FMF products have poor accuracy, which severely limits the retrieval accuracy of the model.

Multiangle polarization observations are a frontier research direction in the field of aerosol remote sensing. These observations have unique advantages in the retrieval of aerosol parameters. Related information analysis work shows that polarization observations can obtain more aerosol information than scalar observations (Chen et al., 2017a;Chen et al., 2017b;Hou et al., 2018). Therefore, the accurate acquisition of more new aerosol parameters based on multiangle polarization observations is of great significance for both atmospheric environmental research and the development of aerosol basic retrieval algorithms.

Although official institutions and some scholars have carried out retrieval studies of aerosol parameters based on multiangle polarization observation platforms, such as POLarization and Directionality of the Earth's Reflectances (POLDER), these studies have their own limitations. For example, the French Laboratoire d'Optique Atmospherique (LOA) only released the main product of fine-mode aerosol optical depth ($AOD_f$) over land (Deuzé et al., 2001;Tanré et al., 2011). Dubovik et al. proposed an optimized retrieval method for polarization observation platforms that can obtain high-precision aerosol optical

parameters (Dubovik et al., 2011). Recently, an operational aerosol product of Generalized Retrieval of Aerosol and Surface Properties (GRASP) based on POLDER data was released (Dubovik et al., 2014), and relevant validation studies show that the product has high retrieval accuracy (Tan et al., 2019;Wei et al., 2020). However, as far as this method is concerned, its computational convergence speed is slow, computational resources are consumed, and a large amount of mathematical statistics is involved. Compared with the traditional lookup table (LUT) method, this method is more difficult to implement.

Although other scholars are conducting related research(Chen et al., 2018;Frouin et al., 2019;Schuster et al., 2019;Li et al., 2020), it is still seldom used in actual engineering applications. In the research of other scholars on the retrieval of new aerosol parameters based on the LUT method, although the results produced by the algorithm have high retrieval accuracy, these studies generally only focus on a specific area, and the spatial scale is not large (Cheng et al., 2012;Xie et al., 2013;Wang et al., 2015;Qie et al., 2015;Wang et al., 2018). There are also fewer studies on the production of long-term aerosol optical

parameter datasets. In 2016, we proposed a method for retrieving FMF based on satellite multiangle scalar and polarization



observations (Zhang et al., 2016), mainly based on multiangle scalar observations to obtain total aerosol optical depth ($AOD_t$), and multiangle polarization observations to obtain $AOD_f$. The ratio of the two is FMF. Compared with the existing MOIDS FMF products, the accuracy of the FMF results obtained by this method is significantly improved, which shows the feasibility of the method. However, there are still some problems that need to be solved if this method is to be applied in large spaces.

For example, the empirical parameters of surface reflectance estimation during scalar retrieval vary greatly with region, and high-precision $AOD_t$ retrieval results can only be obtained in specific regions. In polarization retrieval, there is a problem of low retrieval value for high aerosol loading (Chen et al., 2015;Zhang et al., 2018). In response to these problems, we have also carried out follow-up research work, made certain improvements to the above problems and have achieved more accurate $AOD_t$ and $AOD_f$ in a large space (Zhang et al., 2017;Zhang et al., 2018). Then, in theory, it is possible to achieve the goal of

FMF in a large space. Although Yan et al. achieved high-precision FMF retrieval based on the LUT-SDA method (Yan et al., 2017;Yan et al., 2019), their method is mainly oriented to traditional multispectral scalar sensors. To apply this method to multiangle polarization sensors, it is necessary to perform a series of algorithm adjustments. In previous research, we have achieved high-precision retrieval of $AOD_t$ and $AOD_f$ in a large space. The retrieval method and results can be directly used to obtain FMF, and no more algorithmic adjustments are needed.

This paper is mainly based on the POLDER-3 multiangle polarization sensor on the Polarization and Anisotropy of Reflectances for Atmospheric Science coupled with Observations from a Lidar (PARASOL) satellite and the existing research foundation, and it carried out the retrieval and validation of the FMF in the land area of China. The second chapter of the thesis briefly introduces the FMF retrieval algorithm based on multiangle polarization observation, AErosol RObotic NETwork (AERONET) data and data validation method; the third chapter mainly compares the retrieval results based on the AERONET

ground-based observation data. At the same time, it was also compared with the operational aerosol products of MODIS and GRASP. In Chapter 4, a case study of FMF retrieval is conducted. In this chapter, we also produced a new FMF data set based on the FMF retrieval algorithm of this research. The results of the FMF temporal and spatial distribution over land in China from 2006 to 2013 are obtained. Chapter 5 summarizes the full text and proposes future work prospects.

## 2 Methodology

### 2.1 Introduction to the FMF retrieval method

The technical framework of FMF retrieval in this research is shown in Figure 1. Overall, the FMF retrieval in this study consists of two parts, namely, using the multiangle scalar and polarization data of POLDER-3 to obtain $AOD_t$ and $AOD_f$, and the final ratio of the two is FMF. This method is the same as the retrieval method proposed in our 2016 study (Zhang et al., 2016). However, our previous method is limited by semiempirical parameters on the surface and can only obtain better FMF results

at the urban scale. To obtain stable and accurate results in a large space, we have made major changes to the retrieval methods of $AOD_t$ and $AOD_f$. For the specific retrieval method, please refer to the research we published in 2017 and 2018; here, only a brief introduction is given.



For the retrieval of AOD$_t$, we introduced the empirical orthogonal function (EOF) to estimate the surface reflection contribution under multiangle observations to solve the regional limitation of the semiempirical parameters of the surface in the original

method. Subsequently, this is combined with the retrieval lookup table and substituted into the forward model for simulation calculation, and finally, AOD$_t$ can be obtained through the cost function. The correlation coefficient (r) and root mean square error (RMSE) between the obtained AOD$_t$ and AERONET ground-based observations are 0.891 and 0.097, respectively. For more details about the EOF method, please refer to our 2017 study (Zhang et al., 2017).

For the retrieval of AOD$_f$, our research and other scholars have shown that the AOD$_f$ results obtained by using the official

LOA algorithm have a certain deviation compared with ground-based observations. To improve the retrieval accuracy of AOD$_f$, we proposed the Grouped Residual Error Sorting (GRES) method in 2018 to solve the problem of an inaccurate evaluation function caused by error accumulation under multiangle observation. Based on this method, combined with a bidirectional polarized surface reflectance (BPDF) model to estimate the polarized surface reflectance (Nadal and Bréon, 1999), we have obtained higher-precision AOD$_f$ results in eastern China, and the r and RMSE between the results and the AERONET ground-

based observations are 0.931 and 0.042, respectively. More method details can be found in our research published in 2018 (Zhang et al., 2018).

Based on the new retrieval method, we have obtained higher-precision AOD$_t$ and AOD$_f$ retrieval results on a large spatial scale, which also provides the possibility of obtaining accurate FMF results on a large spatial scale. Next, we will obtain FMF based on the AOD$_t$ and AOD$_f$ retrieved by the new method, validate the FMF retrieval results based on the AERONET ground-based

observation results and further obtain the FMF temporal and spatial distribution results over terrestrial China. Note that since the EOFs during the AOD$_t$ retrieval need to be constructed with the observation results of the POLDER 3*3 window, the resolution of the final FMF retrieval result is also the size of the POLDER 3*3 window (approximately 18 km).

## 2.2 AERONET data

At present, aerosol ground-based products of AERONET have been developed to version V3, and the data of version V2 are

no longer available for download. Among these products, there are two products that can be used to validate the results of satellite FMF retrieval: one is the FMF product based on the spectral deconvolution (SDA) method (O'Neill et al., 2001a;O'Neill et al., 2001b;O'Neill et al., 2003), and the other is based on the size distribution (SD) retrieval product (Dubovik and King, 2000). Generally, SDA products can provide more FMF ground-based results. At present, most base stations in China provide SDA products with level 2.0 data quality. Therefore, SDA products are the first choice for FMF comparison in

this study. However, it is worth pointing out that the Beijing site lacks the SDA product with level 2.0 data quality, so we used the SD product instead. Finally, this study selected the level 2.0 products of 16 AERONET sites in China during 2006-2013 (POLDER on-orbit time) to validate the FMF retrieval results of this study. The specific spatial locations of AERONET sites are shown in Figure 2, and the specific site information is shown in Table 1. However, note that not all AERONET sites have long-term observational data. The sites with long-term observational data are the Beijing, Xianghe, Taihu, and

Hong_Kong_PolyU sites.



The FMF retrieved in this study is the FMF at 550 nm. Neither the SDA product nor the SD product directly provides the FMF result at this wavelength. Therefore, the AERONET FMF needs to be wavelength converted. For SDA products, the products include $AOD_t$, $AOD_f$ at 500 nm and the corresponding Angstrom Exponent (AE), so the FMF of SDA products can be converted to 550 nm by Eq. (1):

$$FMF_{550,SDA} = \frac{\tau_f^{500} \cdot (500/550)^{\alpha_f}}{\tau_t^{500} \cdot (500/550)^{\alpha_t}}, \tag{1}$$

where $FMF_{550,SDA}$ is the FMF of the SDA product at 550 nm after conversion, $\tau_f^{500}$ is the $AOD_f$ at 500 nm, $\tau_t^{500}$ is the $AOD_t$ at 500 nm, $\alpha_f$ is the fine-mode AE, and $\alpha_t$ is the coarse and fine-mode AE.

The SD products provide $AOD_t$ and $AOD_f$ at 440 nm and 675 nm, respectively. Eq. (2)- Eq. (4) can be used to obtain FMF results at 550 nm:

$$\alpha_t = -\frac{ln\,(\tau_t^{675}/\tau_t^{440})}{ln\,(675/440)}, \tag{2}$$

$$\alpha_f = -\frac{ln\,(\tau_f^{675}/\tau_f^{440})}{ln\,(675/440)}, \tag{3}$$

$$FMF_{550,SD} = \frac{\tau_f^{440} \cdot (440/550)^{\alpha_f}}{\tau_t^{440} \cdot (440/550)^{\alpha_t}}, \tag{4}$$

where $FMF_{550,SD}$ is the SD product FMF at 550 nm after conversion, $\tau_f^{675}$ is $AOD_f$ at 675 nm, $\tau_t^{675}$ is $AOD_t$ at 675 nm, $\tau_f^{440}$ is $AOD_f$ at 675 nm, and $\tau_t^{440}$ is $AOD_t$ at 675 nm.

## 2.3 Validation method

In this study, the average value of ground-based observation results within ±30 min of the satellite's transit was used for comparison with the satellite retrieval results. The satellite retrieval result used for comparison is the effective retrieval result centred on the location of the AERONET site within the closest distance in the 3*3 window. The statistical indicators used in the verification include the correlation coefficient (r), mean absolute error (MAE), RMSE, and expected error (EE). The specific statistical evaluation index definitions are shown in Eq. (5)-Eq. (8):

$$r = \frac{Cov(FMF_{retrieval}, FMF_{AERONET})}{\sqrt{D(FMF_{retrieval})}\sqrt{D(FMF_{AERONET})}}, \tag{5}$$

$$MAE = \frac{1}{n}\sum_{i=1}^{n}|FMF_{retrieval} - FMF_{AERONET}|, \tag{6}$$

$$RMSE = \sqrt{\frac{1}{n}\sum_{i=1}^{n}(FMF_{retrieval} - FMF_{AERONET})^2}, \tag{7}$$



$$EE = \pm 0.1 \pm 0.1 \times FMF_{AERONET}, \qquad\qquad (8)$$

where $Cov()$ represents the covariance, $D()$ represents the variance, $FMF_{retrieval}$ represents the FMF retrieval value,
$FMF_{AERONET}$ represents the value of AERONET FMF, and n is the number of validation points.

## 3 Validation and comparison

### 3.1 Introduction to the FMF retrieval method

Figure 3 is a scatter plot of the comparison between the retrieved and AERONET ground-based FMFs. Figures 3(a) to 3(n) list
the verification results at the corresponding sites where the number of matching results is greater than 2. The figure shows that
the FMF results obtained in this study have an overall high correlation with the AERONET ground-based observations. Among
the 14 AERONET sites, r is between 0.508 (Taihu site) and 0.902 (Lanzhou City site). The ranges of MAE and RMSE are
0.096 (Hangzhou_City site) to 0.160 (QOMS_CAS site) and 0.095 (Hangzhou_City site) to 0.184 (QOMS_CAS site). Except
for the QOMS_CAS site, the proportion of results that fell within the EE accounted for approximately 60%. The statistical
indicators of the QOMS_CAS site are all poor. The specific reason is that the site is located at the southern edge of the Qinghai-
Tibet Plateau. It is a high-altitude site and has very little aerosol content. In the AERONET SDA products of 2009-2013, the
5-year average values of $AOD_t$ and $AOD_f$ (500 nm) are only 0.052 and 0.038, respectively. Under the combined influence of
the aerosol model and the surface reflectance estimation error in the retrieval process, it is difficult to accurately retrieve a low
AOD value for satellite observations, resulting in a large deviation of FMF at this site.
We have counted the FMF validation results of different surface types, and the specific information is shown in Table 2. The
r, MAE, and RMSE at all sites in this study are 0.770, 0.143, and 0.170, respectively, and Within EE is 60.96%, again indicating
that the FMF satellite retrieval results of this study are comparable with the ground-based observation results. All the validation
results of this study cover six surface types: urban, barren, grasslands, wetlands, croplands, and forests. Overall, since the
validation data of the barren type mainly come from the QOMS_CAS site, the validation results at this surface type are poor.
Although the r at the other five surface types has a certain change, it is 0.508 (barren)-0.831 (forests)), but in terms of the three
indicators of MAE, RMSE and Within EE, the differences in the five surface types are relatively small, especially Within EE,
which is concentrated at approximately 60%, similar to the site-by-site results. The errors of the FMF retrieval results in this
study are relatively stable at these five surface types.
We further counted the error distribution of the FMF retrieval results, and the statistical results are shown in Figure 4. The
figure shows that the FMF error of this research is mainly distributed between -0.3 and 0.3. This part of the data accounts for
approximately 86%, but the part less than the AERONET ground-based FMF observation value accounts for approximately
75%, indicating that the retrieval result of this study is lower than that of the ground-based observations. The specific reason
needs to be analysed from the FMF retrieval method of this study. The FMF in this study is obtained from the ratio of $AOD_f$
to $AOD_t$, and the retrieval accuracy of the two parameters directly determines the retrieval accuracy of FMF. Therefore, we





compared the retrieved AODs with those of the ground-based data in 2013, and the statistical results are shown in Table 3. The table shows that the mean errors between the $AOD_f$ and $AOD_t$ of our retrieval and the ground-based results are -0.039 and 0.043, respectively, indicating that the $AOD_f$ retrieval result has a negative offset, and the $AOD_t$ retrieval result has a positive offset, that is, the numerator is small and the denominator is large, eventually leading to a small FMF.

## 3.2 Comparison with MODIS products

Because the FMF results obtained by MODIS are different in definition from the ground-based results (Levy et al., 2009), the retrieval results are quite different from the ground-based observation results, which limits the research that depends on the FMF parameter. We compared the retrieved and MODIS FMF with the AERONET ground-based observations to further evaluate the significance of our results. Figure 5 shows the comparison between the two results and the AERONET ground-based observation results from 2011 to 2013. As seen from the figure, compared with ground-based observations, the r of FMF

obtained in this study is 0.812, while that of MODIS is 0.302. The correlation coefficient of the results obtained in this study is much higher than that of MODIS. At the same time, notice that there are many 0 values in the MODIS results. These 0 values are not meaningless but correspond to the situation where there are no fine particle aerosols in the MODIS product definition. Judging from the comparison results, these 0 values have large deviations from the ground-based observation results, and the results of this study are closer to the ground-based observations.

More statistical results of the two are shown in Table 4. The table shows that the FMF results obtained in this study have an MAE of 0.072, an RMSE of 0.102, and a Within EE of 79.72%; the results of MODIS have an MAE of 0.512, RMSE of 0.574, and Within EE of 12.59%. The statistical indicators of the FMF results obtained by our study are greatly improved compared with the MODIS results.

Figure 6 and Figure 7 show the spatial distribution map of the average annual FMF (550 nm) of China in 2013 obtained by

this study and the MODIS product. To facilitate the comparison of the differences in the spatial distribution trends of the two, the two results are normalized, meaning they are divided by the maximum value in the respective FMF image. The figure shows that the results obtained in this study can better reflect the differences in the level of urbanization in China and are more in line with the "Hu Line", reflecting China's population density. That is, in the area to the east of the "Hu Line", the value of the FMF is higher, and the North China Plain, Sichuan-Chongqing Economic Zone, Pearl River Delta, and Yangtze River

Delta are extremely high value areas, while in the area to the west of the "Hu Line", the FMF value is small, the high value area is mainly in the northern Xinjiang region, while the value in the Qinghai-Tibet Plateau is generally low. The results of MODIS are quite different from the results of this study. The MODIS results show that the regions with the highest FMF are Guizhou, Guangxi, Yunnan, and Hainan. The Three Northeast Provinces and the central mountainous areas of Taiwan also have high values. For the North China Plain, Sichuan-Chongqing Economic Zone, and Pearl River Delta, the results are

somewhat similar to this study, while the Yangtze River Delta is a low-value area.





### 3.3 Comparison with GRASP products

In our previous research, the accuracy of FMF calculated from the GRASP product was validated (Wei et al., 2020). The results of comparison with 8 SONET (Sun-sky radiometer Observation NETwork) sites show that the r between GRASP FMF and ground-based observations is 0.77, and Within EE is 62.35%, which is similar to the results of this study in Section 3.1.

However, by comparing the spatial distribution results of the two, we found some differences. We processed the latest V2.06 version of GRASP aerosol products. Figure 8 shows the annual averaged FMF spatial distribution of GRASP in 2013 (also normalized). Compared with Figure 6, we can see certain differences. The relatively high value area of GRASP results is mainly in southern China. We subtracted the results of this study from the average GRASP FMF results and obtained the numerical difference between the two, as shown in Figure 9. The figure shows that the difference between the two in the North

China Plain and the southern Xinjiang region is relatively small. The largest differences are mainly concentrated in the southern and northeastern China and Qinghai-Tibet Plateau regions. The GRASP results in these areas are greater than our results, and a small number of pixels can be larger than 0.3. However, these areas lacked publicly available sunphotometer observations in 2013 and before. The PARASOL ended its exploration mission in October 2013, and it is impossible to compare the subsequent time periods, so it is difficult to directly compare with ground-based observations to illustrate the correctness of

the spatial distribution of the two.

In this study, the ground PM2.5 and PM10 in situ results were compared with the ground-based FMF results. It is expected that the ratio of $PM_{2.5}$ to $PM_{10}$ can be used to analyse the correctness of this study, as well as the GRASP FMF results in the spatial distribution trend. We selected the 2015 Beijing Olympic Sports Center monitoring site (116.407°E, 40.003°N, straight-line distance of less than 4 km), which was the closest to the AERONET Beijing site, and compared the hourly averaged results

of the ratio of $PM_{2.5}$ to $PM_{10}$ with the FMF results. Although the definitions of the two are quite different, the ratio of $PM_{2.5}$ to $PM_{10}$ is actually a parameter of particulate matter near the ground, while FMF is actually a parameter of the atmospheric column of aerosols, but the comparison results of the two (Figure 10) show that there is a correlation between the ratio of $PM_{2.5}$ to $PM_{10}$ and FMF, and the r is 0.709. This result may be because aerosols are mainly distributed near the ground, and $PM_{2.5}$ and $PM_{10}$ can represent different particle modes. In the end, the actual difference between the two parameters is smaller. Since

the ratio of $PM_{2.5}$ to $PM_{10}$ is comparable to the ground-based FMF results, if there are more in situ data, it can indirectly verify the spatial distribution trend of this study and the GRASP results.

Due to the lack of in situ data for particulate matter in China in 2013, this study can only be based on the 2013 environmental protection key city air in the China Statistical Yearbook (http://www.stats.gov.cn/tjsj/ndsj/). The annual average value of air quality is used for limited analysis. We extracted the FMF retrieval results and GRASP results of the corresponding 47 cities

in the statistical yearbook and calculated the annual average FMF of each city for comparison with the ratio of the annual average $PM_{2.5}$ to $PM_{10}$ of each city. The spatial distribution of the administrative regions of these 47 cities is shown in Figure 11. These cities cover most of China's provinces and have a wider spatial distribution range than the AERONET sites in Figure 2. The comparison results in Figure 12 show that although the annual average FMF results of this study in each city are lower





than the annual average results of the ratio of $PM_{2.5}$ to $PM_{10}$, the change trend of the FMF results of this study is better than
the results of GRASP FMF. The r between the FMF of this study and the ratio of $PM_{2.5}$ to $PM_{10}$ is 0.778, while GRASP is
0.472, which can provide evidence for the correctness of the FMF results of this study in the spatial distribution. The low FMF
results in this study are related to the calculation methods of the annual average values of $PM_{2.5}$ and $PM_{10}$ in each city. Generally,
most of the in situ monitoring sites for particulate matter in each city are distributed in urban areas, and the number of sites
distributed in rural areas is small (for example, 9 of the 12 state-controlled sites in Beijing are in urban areas). When calculating
the average FMF of a city, one pixel may contain the results of multiple monitoring stations in place, which makes it difficult
to achieve accurate spatial location matching. To facilitate data processing, all pixels within the urban administrative boundary
are directly used to calculate the average value, and the large number of FMFs in rural areas is generally lower than that in
cities, which ultimately leads to a lower FMF average result.

Based on the validation and comparison results in Sections 3.1 to 3.3, this research has obtained FMF satellite retrieval results
with good accuracy in China, which proves the reliability and stability of the retrieval method. Compared with the MODIS
FMF products, the r, MAE, RMSE and Within EE of the results of this study are all higher than the results of MODIS.
Compared with the GRASP FMF, the results of this study are closer to the results of the ratio of $PM_{2.5}$ to $PM_{10}$ in terms of the
spatial distribution of the entire region of China. The above results all illustrate the effectiveness and advantages of the FMF
retrieval method used in this study. Compared with our original FMF retrieval method, which can only be used at the urban
area scale, this research has achieved FMF retrieval in a large space. Therefore, we will carry out the practical application of
FMF satellite remote sensing retrieval based on the new method.

## 4 FMF retrieval application

### 4.1 Case study

#### 4.1.1 A haze case in North China

Figure 13 contains the retrieval results of a haze pollution incident that occurred in North China on October 5, 2013. The true
colour map shows that North China, especially the Beijing-Tianjin-Hebei region, has several smoke-like pixels, which is a
typical feature of haze pollution in satellite remote sensing images.
Regarding the spatial distribution of $AOD_t$, the $AOD_t$ value (550 nm) in most areas of Beijing-Tianjin-Hebei exceeded 1.0,
and the actual maximum value could reach approximately 2.0. It was the most severely polluted area of the day. There was
also a haze distribution in other surrounding areas, such as eastern Shanxi and northern Shandong, and most areas of Henan
have areas with an $AOD_t$ value of approximately 0.75-1.0. In addition, there are some areas with an $AOD_t$ value of
approximately 0.75 in Anhui and Jiangsu, which fully demonstrates that haze pollution in China is characterized by a large
continuous distribution. In southern China, only the Pearl River Delta and southern Taiwan have areas with a value of





approximately 0.5. The other regions, such as Hubei, Hunan, Jiangxi, Guangdong and Fujian, have low $AOD_t$ values, and most
of the values are concentrated at approximately 0.2, which reflects the cleaner air conditions in South China.

Regarding the spatial distribution of $AOD_f$, the values (550 nm) in most areas of Beijing-Tianjin-Hebei also exceed 1.0, the actual maximum value can reach approximately 1.6, and the overall spatial distribution is similar to the $AOD_t$ distribution. For South China, the value of $AOD_f$ is also relatively small, most of which is concentrated at approximately 0.15, and the values in the Pearl River Delta and Taiwan are larger, with a value of approximately 0.4.

The spatial distribution of FMF is quite different from the distributions of $AOD_t$ and $AOD_f$, reflecting that FMF is another observation dimension in aerosol optical properties. The FMF value (550 nm) in most regions of China is concentrated in the range of 0.6-0.8, and the FMF in most parts of South China also reached this level. This result shows that although the air qualities are quite different, the FMF values are close. It also shows that most of the eastern part of China was dominated by fine-mode aerosols, and only parts of Hubei, Hunan, Jiangxi, and Fujian were dominated by coarse-mode aerosols (with FMF
values of approximately 0.25).

**4.1.2 Case of Sand and Dust in North China**

Figure 14 is the retrieval result of a dust pollution incident in North China on March 9, 2013. The true colour image shows that, except for the Beijing area, which is covered by clouds, the other regions of North China are covered by a large number of brown pixels, which is a typical feature of dust pollution in satellite remote sensing images.

Regarding the spatial distribution of $AOD_t$, the $AOD_t$ value (550 nm) in most areas of the North China Plain is concentrated in the range of 1.2-1.6, and the actual maximum value reaches approximately 2.0, indicating that there is serious dust pollution in most areas of North China. The overall value of Inner Mongolia is relatively small, concentrated at approximately 0.4. Hubei, Anhui, Jiangxi and other regions also have high value areas with $AOD_t$ values of 1.2-1.6, but from the true colour image, these areas are covered by a large number of smoke-like pixels. The high values in these areas are caused by haze pollution; the
$AOD_t$ in South China is generally low, mostly below 0.25.

The spatial distribution of $AOD_f$ shows a different trend from $AOD_t$. The $AOD_f$ (550 nm) of most parts of North China is less than 0.2; Henan, Shandong and other places have areas greater than 0.5; the $AOD_f$ in central China is concentrated between 0.5-1.2, and Hubei has the highest value of 1.2; the overall spatial distribution of South China is similar to that of $AOD_t$, and the value is also low, generally below 0.3.

Regarding the spatial distribution of FMF, the overall trend is again different from the spatial distribution of the $AOD_t$ and $AOD_f$. The FMF value (550 nm) in most areas of North China is concentrated in the range of 0.1-0.2, showing that typical coarse-mode aerosols are dominant; notice that there is a transitional area of FMF at the junction of Hebei, Shandong, and Henan, the value varies between 0.1-0.5, which reflects to a certain extent that these areas are affected by dust and haze, and the composition of aerosols is complex. Central China has a higher FMF value, generally above 0.7, reflecting the dominance
of fine-mode aerosols. Note that in Jiangsu, Guangdong, Fujian and other places, although the values of $AOD_t$ and $AOD_f$ are both low, the FMF values are still high, greater overall than 0.75, showing a strong trend of fine-mode aerosols dominating.





## 4.2 Spatiotemporal distribution results of FMF in China's land area from 2006 to 2013

The retrieval of FMF in all of China's land was performed based on the PARASOL level 1 data from 2006 to 2013. We produced a new FMF dataset of China and obtained the corresponding FMF annual and quarterly average results.

Figure 15 shows the results of the FMF annual average spatial distribution of China from 2006 to 2013. In the results, the FMF spatial change characteristics of China in the 8 years are not obvious; that is, the high value area is always dominated by the area east of the "Hu Line", and the high value area of northern Xinjiang is the most conspicuous in western China. Overall, the FMF in China reached its highest in 2013, but there are certain differences in some regions over time. For example, the value of the North China Plain and Yangtze River Delta region from 2011 to 2013 was higher than that of previous years, and the

value was mainly between 0.6-0.7; the Pearl River Delta region had a higher value from 2007 to 2008, with a value of approximately 0.65. The overall level of the other years is approximately 0.55 and is lower than that of the North China Plain; the Sichuan-Chongqing Economic Zone has a relatively high value in 2013, the FMF value of the entire region is between 0.5-0.7 and is mainly between 0.4-0.6 in the other year; the northern Xinjiang region is similar to the North China Plain region and has a higher value from 2011 to 2013, but the overall level is lower than that of the North China Plain, and the high value

regions are mainly distributed in the economic belt of the northern slope of the Tianshan Mountains, and the value is mainly approximately 0.5.

Figure 16 shows the change in the annual average FMF of China in 2013 compared with 2006. Overall, the annual average value of FMF in China is on the rise, and the provinces with obvious changes in value are mainly Sichuan, Shaanxi, Henan, Hubei, and Yunnan, with an increasing value of up to 0.2. Tibet, Inner Mongolia, Hunan, Jiangxi, and Guangxi have seen

negative changes in the annual average FMF, but the decline is only approximately 0.05, and the FMF in these provinces is still mainly positive.

Figure 17 shows the seasonal average spatial distribution results of FMF in China from 2006 to 2013. In the figure, spring is from March to May, summer is from June to August, autumn is from September to November, and winter is from December to February. As seen from the figure, for the east area of the "Hu Line", the overall FMF reached its highest value in winter,

mainly concentrated in the range of 0.7-0.8; the FMF of southern China still has a relatively high value in the spring, and the overall value is approximately 0.6, while in North China, the plain area is lower, generally between 0.4-0.5; the North China Plain in summer is similar to that in spring, but there is a significant decline in southern China, the value is generally between 0.3-0.5; in autumn, the overall value begins to rise, the value is approximately 0.6. The Sichuan-Chongqing economic zone maintains a relatively high value in all four seasons, and the value in some areas in winter is close to 0.8; the three northeastern

provinces also have high values in winter, and the overall value is between 0.4-0.7. For the area west of the "Hu line", the northern Xinjiang area is higher in autumn and winter, and it can reach 0.7 in some areas in winter, and the southern Xinjiang area also shows a significant increase in winter, with some high values close to 0.6; the Qinghai-Tibet Plateau maintains a low value in all seasons, and the value is mainly concentrated between 0.1-0.3.



## 5 Summary

In this study, the multiangle polarization data of PARASOL were used to perform FMF retrieval, and the retrieval results were compared with the AERONET ground-based observations, MODIS results, and GRASP results. Based on the FMF retrieval method, the retrieval of air pollution cases in China was carried out, and the results of the FMF temporal and spatial distribution in China from 2006 to 2013 were also obtained. Based on the above work content, the conclusions of this research are described as follows:

(1) There is good agreement between the FMF results obtained in this study and the AERONET ground-based observation results. The overall r, MAE, RMSE, and Within EE between the two are 0.770, 0.143, 0.170, and 60.96%, respectively.

(2) The FMF results obtained in this study were more practical than the MODIS FMF products. The r, MAE, RMSE, and Within EE between the FMF results and the ground-based observations are 0.812 versus 0.302, 0.072 versus 0.512, 0.102 versus 0.574, 79.72% versus 12.59%, respectively.

(3) Compared with the GRASP FMF, the FMF results obtained in this study are closer to the ratio of $PM_{2.5}$ to $PM_{10}$ in terms of the spatial distribution trend. Compared with the annual average ratio of $PM_{2.5}$ to $PM_{10}$ in 47 Chinese cities in 2013, the r of this study is 0.778, and GRASP is 0.472.

(4) According to the annual and quarterly average FMF results in China from 2006 to 2013, the spatial distribution trend of China's FMF does not change significantly with the year, and the high-value area is mainly maintained in the area east of the

"Hu Line". The FMF showed an increasing trend in 2013 compared with that of 2006. The FMF in China has the highest value in winter and the lowest value in summer. The Sichuan-Chongqing economic zone has a relatively high FMF value in all four seasons.

The FMF retrieval method in this study has significance for the development of aerosol polarization satellite remote sensing algorithms, and the FMF results obtained in China also have good practical value for application research in the field of

atmospheric environments. China has launched the Gaofen-5 (GF-5) satellite equipped with a new multiangle polarization sensor. With the release of GF-5 satellite data in the future, the results of this study can also provide algorithmic support for the application of its multiangle polarization sensor in the field of atmospheric environmental monitoring and are expected to produce subsequent FMF datasets. However, there are some shortcomings in this research. For example, the retrieval of FMF still depends on the accuracy of the two parameters $AOD_f$ and $AOD_t$. In our previous research, although higher-precision

results of $AOD_f$ and $AOD_t$ have been obtained, the FMF error is related to the error of the two retrieval parameters. The transmission of the error will eventually amplify the retrieval error of FMF. Compared with the individual retrieval of $AOD_f$ and $AOD_t$, the retrieval of FMF is still difficult. In the future, it is still necessary to further improve the retrieval accuracy of $AOD_f$ and $AOD_t$. In addition, due to the limitation of the validation data, we are temporarily unable to further discuss the correctness of the spatial distribution trend of the FMF in this study and GRASP, and only the results of the ratio of $PM_{2.5}$ to

$PM_{10}$ were used for indirect comparison. In the future, we can try to perform FMF retrieval in other regions with many ground-based observations around the world to further compare the findings of the two results.



*Data availability.* The FMF datasets produced in this study can be requested from the corresponding author(lizq@radi.ac.cn).

*Author contributions.* ZL conceived and designed the study. YZ and LQ collected and processed the remote sensing data. YZ and YW performed the FMF retrievals. YZ and YX compared the retrieval results with the AERONET, MODIS, GRASP products. WH and LL analyzed the spatiotemporal trends of FMF in China. ZL and YW collected and processed the in situ data. YZ and ZL prepared the paper with contributions from all coauthors.

*Competing interests.* The authors declare that they have no conflict of interest.

*Acknowledgments.* This work was supported by the National Natural Science Fund of China (41901294), the National Natural Science Foundation of Chongqing, China (cstc2019jcyj-msxm0726), the Science and Technology Department of Sichuan Province Foundation (2019YFS0470), the Chengdu Science and technology project (2018-ZM01-00037-SN).

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



Table 1. AERONET site information employed in this study. The land cover types are from the MODIS MCD12 landcover product.

| AERONET sites | Longitude (°E) | Latitude (°N) | Land cover type |
|---|---|---|---|
| Beijing | 116.381 | 39.977 | Urban |
| Hangzhou_City | 120.157 | 30.290 | Urban |
| Hefei | 117.162 | 31.905 | Urban |
| Hong_Kong_PolyU | 114.180 | 22.303 | Urban |
| Kaiping | 112.539 | 22.315 | Urban |
| Lanzhou_City | 103.853 | 36.048 | Urban |
| Minqin | 102.959 | 38.607 | Barren |
| NAM_CO | 90.962 | 30.773 | Grasslands |
| NUIST | 118.717 | 32.206 | Urban |
| QOMS_CAS | 86.948 | 28.365 | Barren |
| SACOL | 104.137 | 35.946 | Grasslands |
| Taihu | 120.215 | 31.421 | Wetlands |
| Taipei_CWB | 121.538 | 25.015 | Urban |
| Xianghe | 116.962 | 39.754 | Croplands |
| Xinglong | 117.578 | 40.396 | Forests |
| Zhongshan_Univ | 113.390 | 23.060 | Urban |




Table 2. **FMF validation results of different surface types**

| Land cover type | N | r | MAE | RMSE | Within EE |
|---|---|---|---|---|---|
| Overall result | 1186 | 0.770 | 0.143 | 0.170 | 60.96% |
| Urban | 421 | 0.733 | 0.139 | 0.163 | 66.03% |
| Barren | 63 | 0.711 | 0.158 | 0.182 | 42.86% |
| Grasslands | 113 | 0.777 | 0.137 | 0.170 | 61.06% |
| Wetlands | 150 | 0.508 | 0.145 | 0.176 | 62.00% |
| Croplands | 394 | 0.651 | 0.146 | 0.174 | 57.86% |
| Forests | 45 | 0.831 | 0.133 | 0.159 | 66.66% |






**Table 3. Statistical analysis of AOD$_f$ and AOD$_t$ errors**

| Retrieval parameter | Mean error | Proportion of negative offset | Proportion of positive offset |
|---|---|---|---|
| AOD$_f$ (550 nm) | -0.039 | 61.96% | 38.04% |
| AOD$_t$ (550 nm) | 0.043 | 44.57% | 55.43% |
| FMF (550 nm) | -0.078 | 68.47% | 31.53% |



**Table 4 Comparison between the retrieved and MODIS FMF**

| Retrieval parameter | MAE (this study) | RMSE (this study) | Within EE (this study) | MAE (MODIS) | RMSE (MODIS) | Within EE (MODIS) |
|---|---|---|---|---|---|---|
| FMF (550 nm) | 0.072 | 0.102 | 79.72% | 0.512 | 0.574 | 12.59% |




**Figure 1. FMF retrieval technology framework of this research**



**Figure 2. The spatial distribution of AERONET sites selected in this study**

















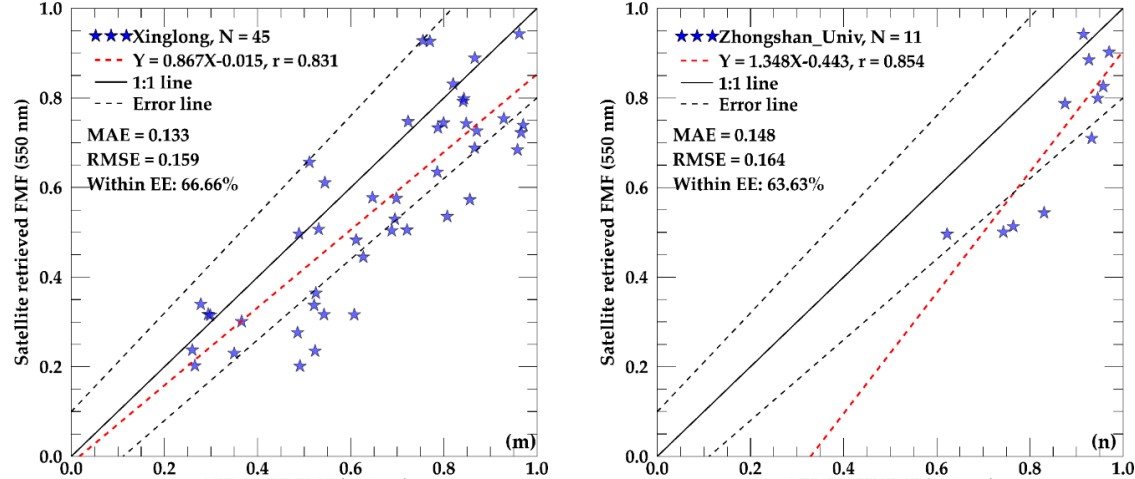

**Figure 3. FMF results comparison at 14 AERONET sites. (a) - (d) are the validation results for the Beijing, Hangzhou_city, Hongkong_PolyU, Kaiping, Lanzhou_city, NAM_CO, NUIST, QOMS_CAS, SACOL, Taihu, Taipei, Xianghe, Xinglong, Zhongshan_Univ sites, respectively.**






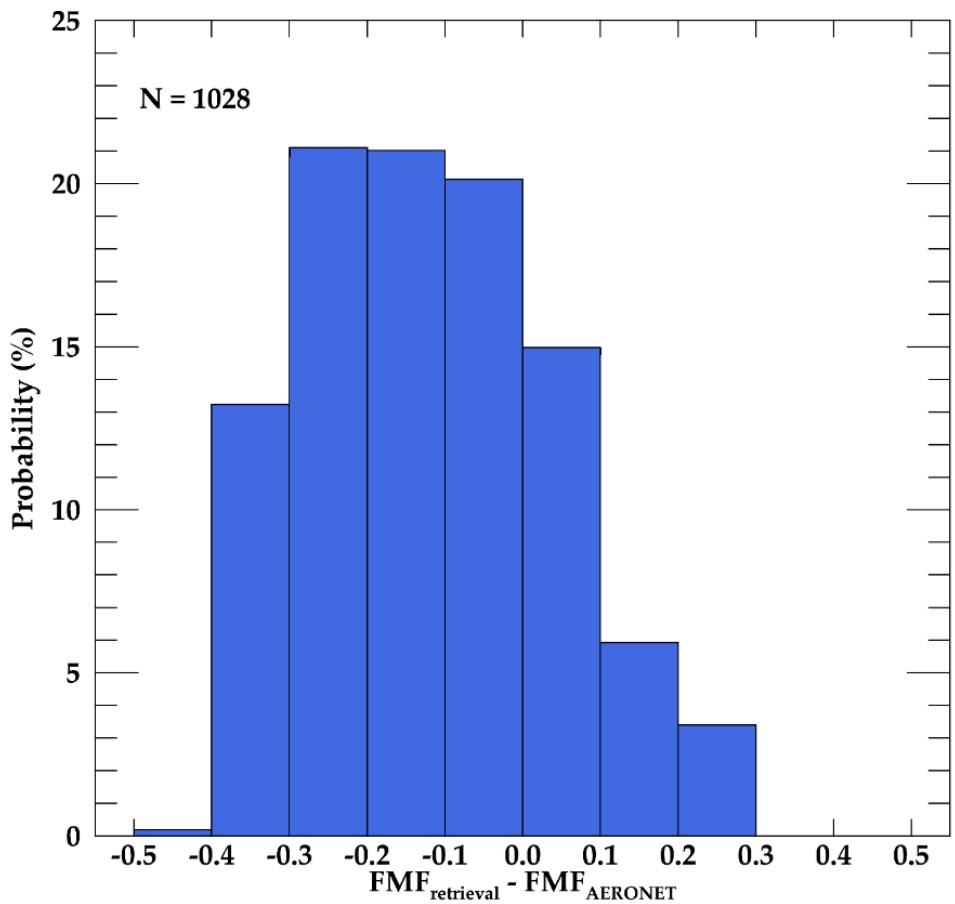

**Figure 4. FMF retrieval error distribution results**



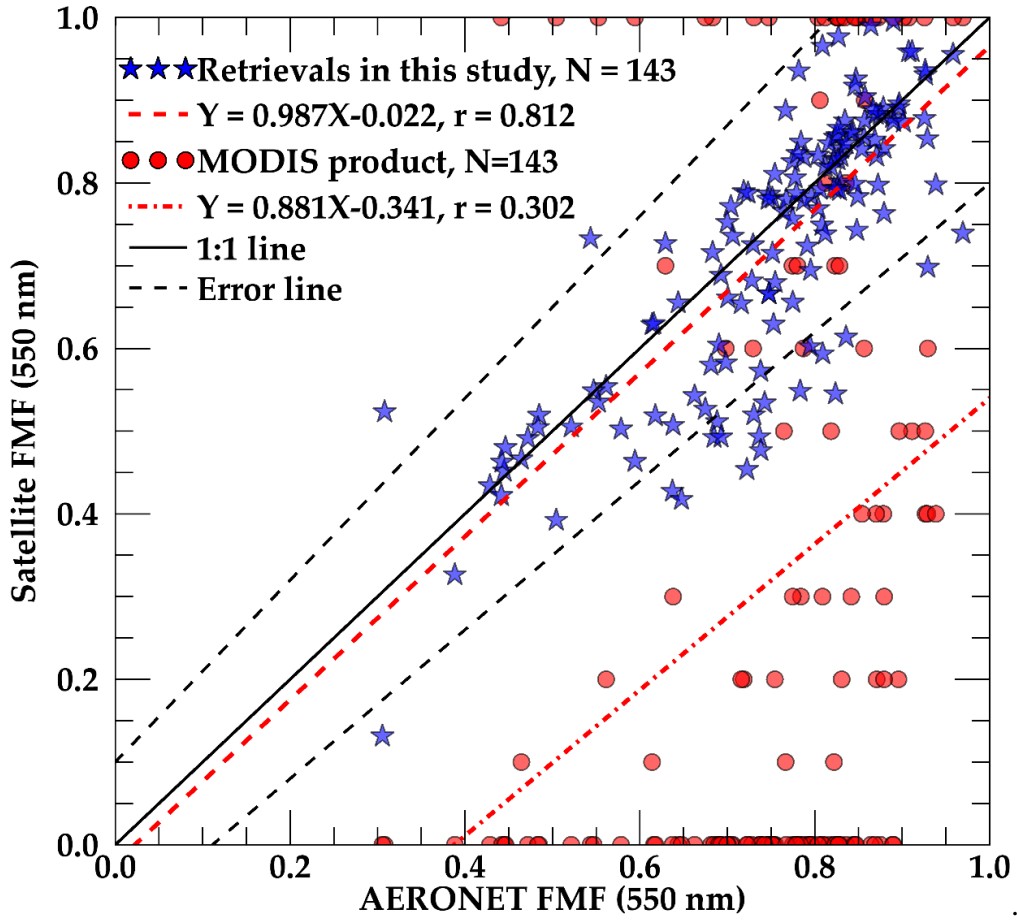

**Figure 5. Comparison between the results of this study and MODIS FMF with AERONET**





**Figure 6. Distribution of normalized FMF of China in 2013 (results of this study)**




**Figure 7. Distribution of normalized FMF of China in 2013 (MODIS results)**

**Figure 8. Distribution of normalized FMF of China in 2013 (GRASP results)**




**Figure 9. Differences in FMF results in China in 2013 (GRASP results minus the retrieved results)**



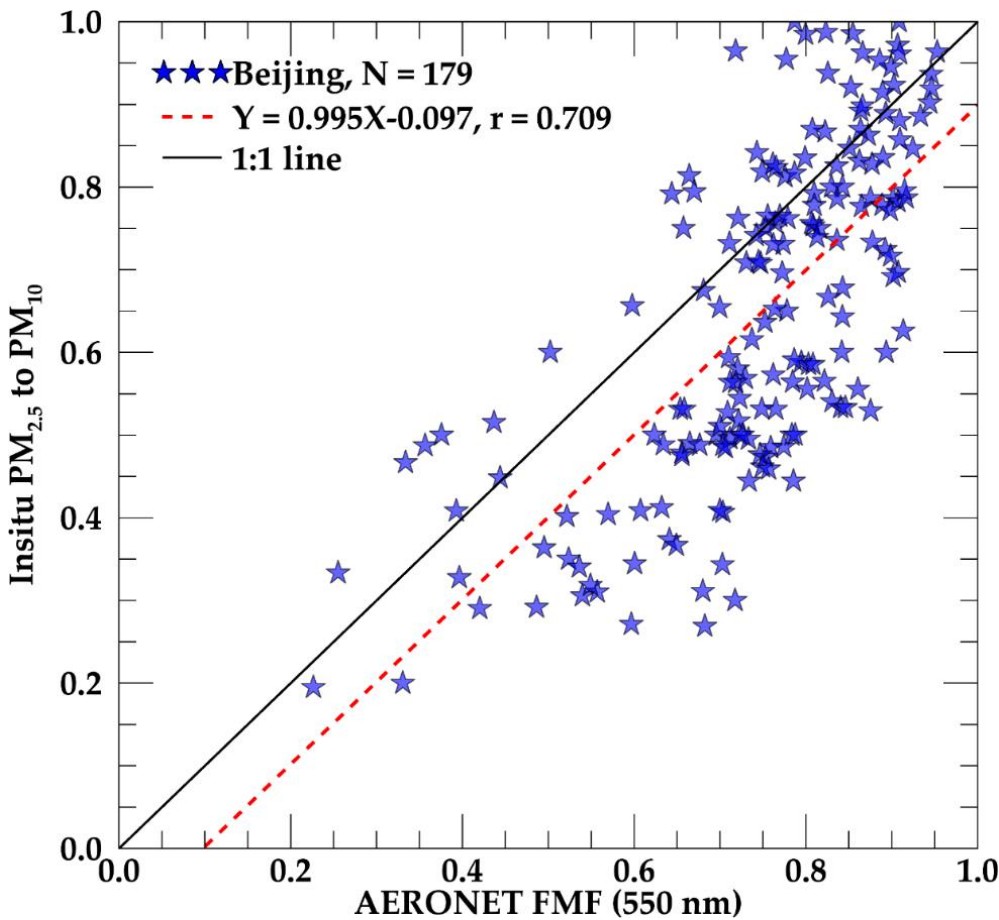

**Figure 10. Comparison between the ratio of PM2.5 to PM10 and FMF (hourly average)**


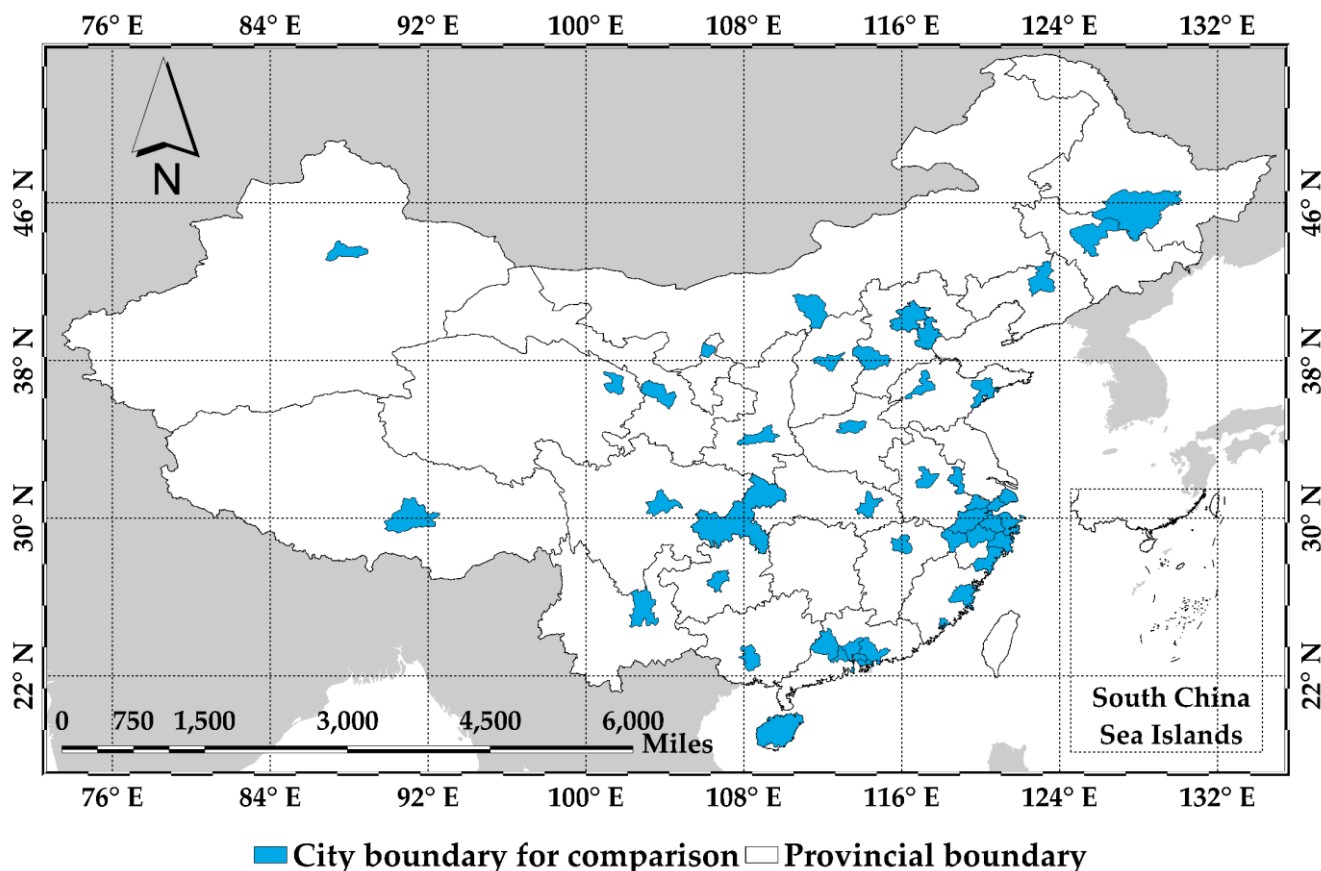

**Figure 11. Forty-seven urban administrative regions in China used to compare the annual average FMF**





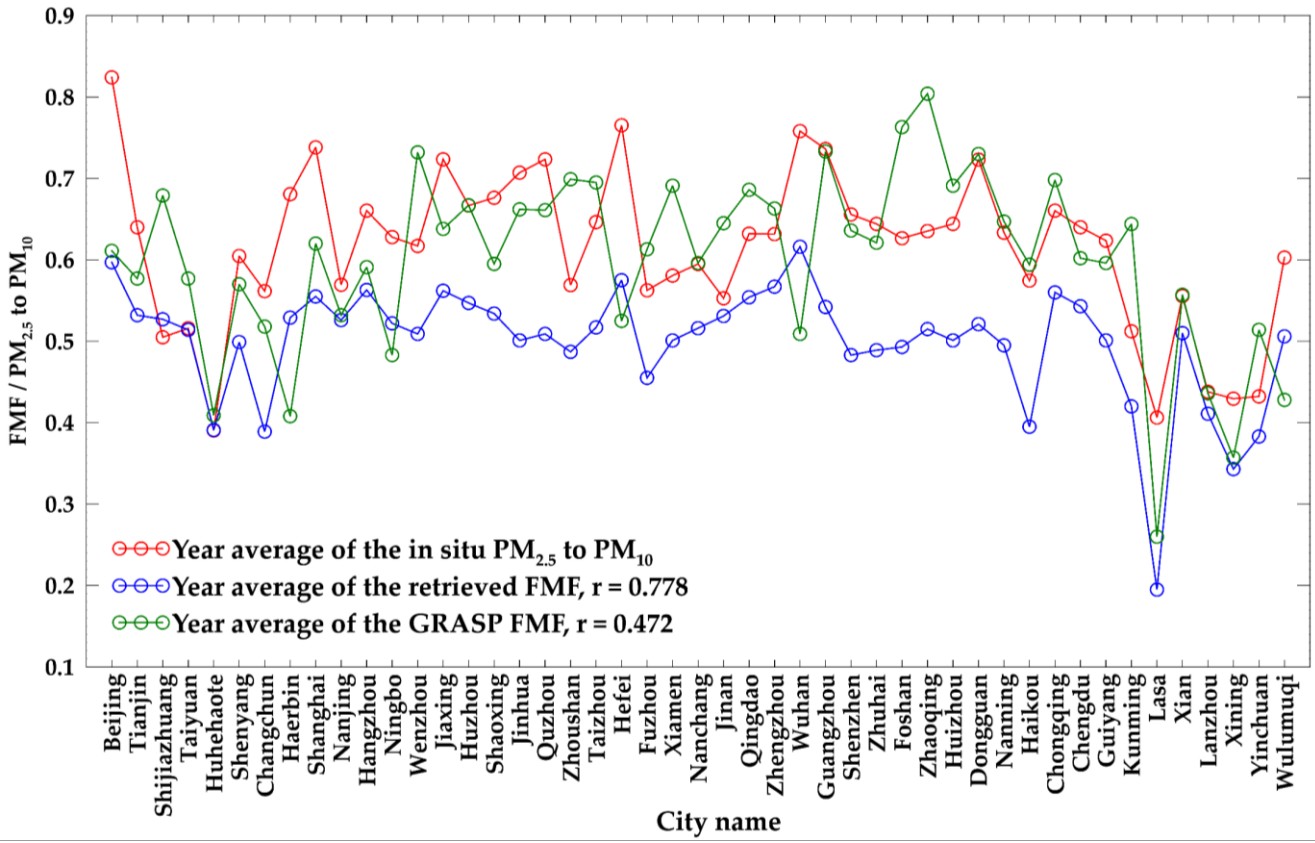


**Figure 12. Comparison of the results of the retrieved and GRASP FMF with the urban average of the ratio of PM$_{2.5}$ to PM$_{10}$ (2013)**



**Figure 13. The retrieval result of the haze case in North China on October 5, 2013.**
**(a) is the true colour image of POLDER, (b) is the retrieval result of the $AOD_t$, (c)**
**is the retrieval result of the $AOD_f$, and (d) is the retrieval result of the FMF.**



**Figure 14. Same as in Figure 13 but for October 5, 2013.**







**Figure 15. The results of the FMF annual average spatial distribution of China. (a)-(h) are the results from 2006 to 2013, respectively.**





**Figure 16. Numerical distribution of the spatial variation in FMF in China (2013 minus 2006)**






**Figure 17. Results of the FMF seasonal average spatial distribution of China. (a)-(d) are the results of spring, summer, autumn and winter from 2006 to 2013.**