# Peer review of "Retrieval of Aerosol Fine-mode Fraction over China from Satellite Multiangle Polarized Observations: Validation and Comparison"

_Atmospheric Measurement Techniques, 2020_

## Referee Comment (RC1) · Anonymous Referee #1 · 27 Oct 2020

This is an interesting study looking at the fine mode fraction (FMF) of aerosol optical depth in China. The main focus is the comparison between results from a previously-published algorithm of the authors applied to POLDER data (referred to in the paper as "the algorithm"), the GRASP retrieval applied to POLDER, the MODIS Dark Target land retrieval, and the AERONET spectral deconvolution algorithm (SDA) and almucantar scan size distribution (SD) retrieval algorithm. The topic is relevant to the journal and the Special Issue. After total AOD, fine vs. coarse AOD is one of the next main frontiers of interest.

The quality of language needs some improvement. I appreciate the authors' first language is not English, and they have done a good job explaining what was done in the analysis. But some copy-editing will be necessary to bring the article to publication standards as phrasing is strange in places (too many to go through as a reviewer). This might be able to be handled by the journal production office, but if the authors have access to a service or colleague who is able to give a proof-read that would be beneficial as well. Again, the authors have done a pretty good job with the writing overall.

My recommendation is for major revisions, and I would like to review the revised version.

Major comments:

1. It is not just the FMF which is of interest, but the overall fine and coarse AOD. It seems like a missed opportunity not to evaluate e.g. the fine mode and total AOD as well. Looking at only FMF we don't know if a bias in that is because the retrievals have errors in the total AOD or just the ratio between modes. This is briefly shown (Table 3) but only via summary metrics (would be good to see the data points) and only for the authors' approach (not MODIS or GRASP). I recommend the authors add this in the revised version. These could be also split by, for example, aerosol type or surface type (as these are some factors which often affect retrieval performance).

2. The authors' interpretation of MODIS Dark Target land FMF is, to my knowledge, not correct. The MODIS land fine weighting parameter eta is not a "fine mode fraction" but a "fine model fraction". The Dark Target land retrieval mixes between two bimodal size distributions, one which is mostly fine mode and one which is mostly coarse mode. See Levy 2007 (https://agupubs.onlinelibrary.wiley.com/doi/full/10.1029/2006JD007815), especially Figure 4 there. So in MODIS, FMF=0 does not mean no fine mode aerosol. It means that the proportion of the bimodal fine-dominated optical model is zero. There is still some fine mode aerosol from the coarse-dominated optical model. Similarly FMF=1 in MODIS still has some coarse aerosol present. This misinterpretation affects all the discussion of MODIS results.

3. The definition of FMF in other products is not be the same either, e.g. the AERONET SDA assumes a combination of fine and coarse modes but the AERONET sky-scan retrieval and (I think) GRASP look for a minimum in the size distribution and make the fine/coarse split there. The manuscript should be more detailed about the exact definition of fine mode fraction within the products, and make sure they are comparable. If not then the discrepancies will be partly due to definition differences rather than retrieval problems.

Minor comments:

1. Line 98: is the EOF method similar to the MISR land approach? That could be mentioned (and compared if different) as it is likely that the readership of this journal would have some familiarity with it.

2. Line 133: Angstrom should be written Ångström here.

3. Line 154: I don't know why it makes sense for EE to be +/-(0.1+10%). Why should FMF uncertainty depend on FMF? Is high FMF slightly harder to retrieve? More justification is needed here. If this was used in a previous study, we need to see the justification there, and if there wasn't one, then that's an issue. I do not see a physical reasoning why FMF uncertainty should be a function of FMF.

4. Line 154: Also, when calculating this metric, the uncertainty on AERONET FMF should be accounted for as well (this is dependent on AOD but can also be of order 0.1 to 0.15: this is discussed in some AERONET publications). Unlike AERONET total AOD, AERONET FMF cannot be considered a reference truth because there are non-negligible uncertainties in both the AERONET SDA and SD retrievals.

5. Lines 220-221: I believe the latest GRASP is version 2.1, not 2.0.6 as stated here. Also, which GRASP product? There are 3 separate GRASP POLDER data sets with different assumptions about aerosol size distribution form. See this paper by Chen et al for more information: https://essd.copernicus.org/preprints/essd-2020-224/ More

information should be added to the manuscript, and if possible the analysis should use the latest GRASP version. Including those results in this paper (rather than just citing an older evaluation study) would also help to compare GRASP and the authors' new approach. Right now it is still not clear to me which is better or what the relative benefits are.

6. Line 243: if possible, more information about the PM2.5 and PM10 surface measurements should be made here. For example is this BAM, filter, or something else?

7. Section 4.1: I do not see much value in showing these two case studies. It's just a couple of maps and text describing them. There isn't really enough context or external data sets brought in to make them interesting. I recommend removing section 4.1 (so section 4.2 would just be called section 4), unless the authors can provide additional material of scientific interest to make the reader care about these examples.

8. Section 4.2: It would be good to add AOD maps here as well, for additional context.

9. Figure 1: "Fuction" should say "Function".

10. Figure 2: I don't know why there is a color bar (FMF) on this figure, given it is only showing site locations on a blank background map.

11. Figure 3: These panels should have separate titles or similar (e.g. a, b) to separate them. Also, I would remove the regression lines. I don't think they add anything, and don't think they are appropriate. In some cases the relationships aren't linear (e.g. QOMS_CAS), and the technique is not valid because (1) it is not accounting for uncertainty in the AERONET reference data and (2) the data are constrained by the possible bounds of FMF (i.e. 0-1) meaning that errors cannot be Gaussian and unbiased. Both of these means that the assumptions required for validation are not satisfied.

12. Figure 4: It would be good to show a second histogram in addition, filtered for points where the AOD is above a certain value (e.g. 0.2?). We would expect that this

would be thinner because the sensitivity to FMF should be better when AOD is high. So this would be interesting to see how the width and midpoint of the distribution change.

13. Figure 5: same comment about regression line as for Figure 3. Also, both data sets have 143 points here: is this for the points where MODIS and POLDER are all matched together? This should be stated in the paper, this was not clear to me.

14. Figures 6-8: can we have more than 3 points labeled on the color scale? Also, it would be clearer to combine these together into one 1 figure, possibly with figure 9 as well.

15. Figure 9: is this FMF difference or normalized FMF difference? This was not clear to me.

16. Figure 10: same comment about regression line as for Figure 3.

17. Figures 15, 16: I don't see much value in these figures and suggest deleting them. The overall spatial distribution looks fairly similar year to year, and I think the seasonal maps in Figure 17 are more useful. Likewise, I am not convinced that the differences in Figure 16 are realistic. It looks like a FMF difference of +0.2 across many parts of China, even remote areas where the aerosol is mostly dust. So in my view it could easily be a calibration drift, as POLDER had no on-board calibration. There is only a paragraph devoted to Figure 16 anyway. If the authors wish to discuss trends, it would make more sense to show also total AOD and fine mode AOD (so we can see which is increasing) and bring in some additional satellite, model, or ground-based data to help verify and understand the mechanisms. If Figures 15 and 16 are removed, then Figure 17 could also be moved earlier in the manuscript, close to where the authors' retrieval method is introduced.

---

## Referee Comment (RC2) · Anonymous Referee #2 · 27 Oct 2020

GENERAL COMMENTS

This manuscript by Zhang et al. conducted the fine mode fraction (FMF) retrieval from muliti-angular polarimeter (PARASOL). Technically, the total AOD is determined from intensity measurements, and fine mode AOD is derived from muliti-angular polarized measurements. Then the ratio of AOD and fine mode AOD derives FMF. This method generally sounds, and has been published in Zhang et al. (2017, 2018).

This manuscript is mainly focus on the validation of retrieved FMF using AERONET, MODIS, PARASOL/GRASP products. The main concern here is that each product may have different definition of their FMF, this should be fully considered before conducting

validation and inter-comparison. For example, MODIS FMF over land is the ratio to reflectance instead of total AOD; therefore MODIS FMF over land has little physical meaning. Over ocean, by single scattering approximation, FMF can be approximated as weighted for AOD (see discussions in Remer et al., 2005). Additionally, the objective is not clear why the authors pay close attention to FMF instead of fine mode AOD, the uncertainties in both AOD and fine AOD could significantly worsen the FMF quality, and a good FMF doesn't necessarily produce a good estimation of fine mode AOD. Overall, I think this manuscript is within the scope of AMT. Some comments and concerns are required to be addressed and clearly stated before being published. The specific comments are listed as follow.

SPECIFIC COMMENTS

Line 39: please be cautious to interpret MODIS FMF over land, it is weighted of reflectance instead of AOD (see discussions in Remer et al., 2005; Chen et al., 2020);

Line 53: This is not true. Please check Chen et al., 2020 (10.5194/essd-2020-224)

Line 71-72: 'there is a problem of low retrieval value for high aerosol loading' ??? Could you specify it, underestimation for high AOD or FMF?

Line 82: thesis?? -> study

Line 148: 3x3 window ? is it equivalent to 3x18km?

Line 154: is there any intention or reference to use $\pm0.1\pm10\%$ EE for FMF?

Line 158: Section name is wrong.

Figure 3: is this all points from 2006-2013? Any filter scheme used, please clarify.

Line 177: errors . . . are stable. . . ?? please consider 'uncertainty'

Line 182: the definitions of AERONET FMF and retrieved AODf/AOD are not identical

Table 3: Number of points is critical, as well as other parameters (r, rmse, etc.)

[Figure]

Line 220: Please identify products name and version, and last access, etc. (This is necessary for all products used in the manuscript)

Line 222: what do you mean normalized FMF?

Section 3.3: why only 2013 data is compared? It would be interesting to check more data 2006-2013 and other related parameters, e.g. AOD and fine mode AOD, to make the conclusion more solid.

Figures 6, 7, 8: it is important to mention the spatial resolution, visually, the derived FMF in figure 6 has much coarser resolution than others

Figures9, 16: the quality of figures showing differences can be improved by using more adequate colorbar

Line 346: throughout the manuscript, no place specified the MODIS (TERRA or AQUA or both) dataset

Line 370: Is there any specific reason to pay close attention to FMF instead of fine mode AOD? On one hand, the uncertainties in both AOD and fine AOD could significantly worsen the FMF, on the other hand, a good FMF doesn't necessarily produce a good estimation of fine mode AOD, which can compensate by AOD and fine AOD, right?

---

## Author Comment (AC2) · 30 Dec 2020

Thanks for your helpful comments, we have revised the paper based on your comments. The following is a one-to-one response to your comments.

**GENERAL COMMENTS**

This manuscript by Zhang et al. conducted the fine mode fraction (FMF) retrieval from mulitiangular polarimeter (PARASOL). Technically, the total AOD is determined from intensity measurements, and fine mode AOD is derived from multi-angular polarized measurements. Then the ratio of AOD and fine mode AOD derives FMF. This method generally sounds, and has been published in Zhang et al. (2017, 2018). This manuscript is mainly focus on the validation of retrieved FMF using AERONET, MODIS, PARASOL/GRASP products. The main concern here is that each product may have different definition of their FMF, this should be fully considered before conducting validation and inter-comparison. For example, MODIS FMF over land is the ratio to reflectance instead of total AOD; therefore MODIS FMF over land has little physical meaning. Over ocean, by single scattering approximation, FMF can be approximated as weighted for AOD (see discussions in Remer et al., 2005). Additionally, the objective is not clear why the authors pay close attention to FMF instead of fine mode AOD, the uncertainties in both AOD and fine AOD could significantly worsen the FMF quality, and a good FMF doesn't necessarily produce a good estimation of fine mode AOD. Overall, I think this manuscript is within the scope of AMT. Some comments and concerns are required to be addressed and clearly stated before being published. The specific comments are listed as follow.

In the revised paper, we have discussed the differences in the definition of different FMF products. Please check our revised paper later. This part is also included in our answer to your comment below. In 2015, we proposed the PMRS model (Zhang et al., 2015), which is a model based on physical methods to estimate PM2.5 concentration. In that model, FMF is an important input parameter and cannot be replaced by AODf. Since the existing MODIS FMF products are difficult to meet the application requirements of the PMRS model, we started the research of using multi-angle polarization sensors to retrieve FMF. In addition, FMF can also be used to distinguish anthropogenic and natural aerosol types (Bellouin et al., 2005). We think that FMF is also important for research in the field of atmospheric environment.

**References:**

Zhang, Y., and Li, Z.: Remote sensing of atmospheric fine particulate matter (PM2.5) mass concentration near the ground from satellite observation, Remote Sensing of Environment, 160, 252-262, 10.1016/j.rse.2015.02.005, 2015.

Bellouin, N., Boucher, O., Haywood, J., Reddy, M.S., 2005. Global estimates of aerosol direct radiative forcing from satellite measurements. Nature 438, 1138–1141.

**SPECIFIC COMMENTS**

Line 39: please be cautious to interpret MODIS FMF over land, it is weighted of reflectance instead of AOD (see discussions in Remer et al., 2005; Chen et al., 2020);

**Answer:** After we read the comments of you and another reviewer, we realized that we had a misunderstanding of MODIS FMF. We have rewritten this paragraph as follows:

However, other new aerosol optical parameters, such as the fine-mode fraction (FMF), are quite different in definition from the ground-based observations (Remer et al., 2005;Levy et al., 2010), which makes them incomparable.

**References:**

Remer, L. A., Kaufman, Y. J., Tanré, D., Mattoo, S., Chu, D. A., Martins, J. V., Li, R. R., Ichoku, C., Levy, R. C., and Kleidman, R. G.: The MODIS Aerosol Algorithm, Products, and Validation, Journal of the Atmospheric Sciences, 62, 947-973, 2005.

Levy, R. C., Remer, L. A., Kleidman, R. G., and Mattoo, S.: Global evaluation of the Collection 5 MODIS dark-target aerosol products over land, Atmospheric Chemistry & Physics, 10, 10399-10420, 2010.

Line 53: This is not true. Please check Chen et al., 2020 (10.5194/essd-2020-224).

**Answer:** Our expression was not clear. We wanted to say that LOA only provides  $AOD_f$  in its operational aerosol products over land. Chen et al. also mentioned this information in their section 4.1 (10.5194/essd-2020-224). We have rewritten this paragraph as follows:

For example, the French Laboratoire d'Optique Atmospherique (LOA) only provided the fine-mode aerosol optical depth (AODf) datasets in its operational product over land (Deuzé et al., 2001; Tanré et al., 2011), the total aerosol optical depth (AODt) was not provided (Chen et al., 2020).

**References:**

Deuzé, J. L., Bréon, F. M., Devaux, C., Goloub, P., Herman, M., Lafrance, B., Maignan, F., Marchand, A., Nadal, F., Perry, G., and Tanré, D.: Remote sensing of aerosols over land surfaces from POLDER-ADEOS-1 polarized measurements, Journal of Geophysical Research, 106, 4913, 10.1029/2000jd900364, 2001.

Tanré, D., Bréon, F. M., Deuzé, J. L., Dubovik, O., Ducos, F., François, P., Goloub, P., Herman, M., Lifermann, A., and Waquet, F.: Remote sensing of aerosols by using polarized, directional and spectral measurements within the A-Train: the PARASOL mission, Atmospheric Measurement Techniques, 4, 1383-1395, 10.5194/amt-4-1383-2011, 2011.

Chen, C., Dubovik, O., Fuertes, D., Litvinov, P., Lapyonok, T., Lopatin, A., Ducos, F., Derimian, Y., Herman, M., Tanré, D., Remer, L. A., Lyapustin, A., Sayer, A. M., Levy, R. C., Hsu, N. C., Descloitres, J., Li, L., Torres, B., Karol, Y., Herrera, M., Herreras, M., Aspetsberger, M., Wanzenboeck, M., Bindreiter, L., Marth, D., Hangler, A., and Federspiel, C.: Validation of GRASP algorithm product from POLDER/PARASOL data and assessment of multi-angular polarimetry potential for aerosol monitoring, Earth Syst. Sci. Data Discuss., 2020, 1-108, 10.5194/essd-2020-224, 2020.

Line 71-72: 'there is a problem of low retrieval value for high aerosol loading'??? Could you specify it, underestimation for high AOD or FMF?

**Answer:** The underestimation is for AODf for high aerosol loading. We have rewritten this sentence as follows:

In polarization retrieval, the problem of a low AODf retrieval value for high aerosol loading exists

Line 82: thesis?? -> study. Answer: We have corrected it.

Line 148: 3x3 window ? is it equivalent to 3x18km?

**Answer:** Yes, it is equivalent to 3x18km, which is about 54 km. We have added this information as follows:

The satellite retrieval result used for comparison is the effective retrieval result centred on the location of the AERONET site within the closest distance in the 3\*3 window (about 54 km).

**Line 154: is there any intention or reference to use $\pm 0.1 \pm 10\%$ EE for FMF?**

Answer: The other reviewer also mentioned this issue. However, there does not seem to be a unified standard for EE definition of FMF, different studies have different standards. For example, the study of Cheng et al. did not define the EE of FMF. The study of Yan et al. defined the EE of FMF as  $\pm 0.4$ . The study of Chen et al. defines three types of FMF EE: +/-(0 + 40%), +/-(0 + 25%), +/-(0.03 + 20%). We have reconsidered the definition of EE for FMF. Firstly, we believe that the EE of FMF should not increase as the value increases, which is different from AOD. Secondly, the ground-based FMF has a certain error. According to the research of O'Neill et al., the SDA method has an uncertainty of about 0.1. We considered the absolute error part (0.1) of the previous EE of FMF and the uncertainty (0.1) of the ground-based FMF, and finally changed the EE of FMF in this study to  $\pm 0.2$ .

**References:**

T, Cheng, X, et al. Aerosol optical depth and fine-mode fraction retrieval over East Asia using multiangular total and polarized remote sensing[J]. Atmospheric Measurement Techniques, 2012.

Yan X, Li Z, Shi W, et al. An improved algorithm for retrieving the fine-mode fraction of aerosol optical thickness, part 1: Algorithm development[J]. Remote Sensing of Environment, 2017, 192:87-97.

Chen, X., de Leeuw, G., Arola, A., Liu, S., Liu, Y., Li, Z., and Zhang, K.: Joint retrieval of the aerosol fine mode fraction and optical depth using MODIS spectral reflectance over northern and eastern China: Artificial neural network method, Remote Sensing of Environment, 249, 112006, 2020

O'Neill, Norm T, Dubovik, et al. Modified Ångström Exponent for the Characterization of Submicrometer Aerosols[J]. Applied Optics, 2001.

Line 158: Section name is wrong.

Answer: We have modified the section name as 'Validation against AERONET ground-based data'.

Figure 3: is this all points from 2006-2013? Any filter scheme used, please clarify.

**Answer:** Yes, this is all the matched points from 2006 to 2013. When the retrieved  $AOD_f$  is greater than the retrieved  $AOD_t$ , we consider this situation as a failure of the FMF retrieval, and the results of this part were not involved in the comparison. These results account for about 10% We have added those information in section 2.3 as follows:

Note that when the retrieved  $AOD_f$  is greater than the retrieved  $AOD_b$ , we consider this situation as a failure of the FMF retrieval, and the results of this part were not involved in the comparison. These results account for about 10%.

Line 177: errors : : : are stable: : : ?? please consider 'uncertainty'. **Answer:** We have corrected it.

Line 182: the definitions of AERONET FMF and retrieved AODf/AOD are not identical.

**Answer:** We agree that the definitions of the AERONET FMF and retrieved FMF are not identical. However, they have some similarities. The definition of the  $AOD_f$  in our study is indefinite, and it has no clear cut-off particle size. Similarly, there is also no clear definition of  $AOD_f$  in the groundbased SDA algorithm. Therefore, we think that although they are not equivalent, the two are comparable. We prefer to use the SDA FMF as the 'truth value' for validation.

Table 3: Number of points is critical, as well as other parameters (r, rmse, etc.).

**Answer:** We have added the information of the number of points, r and bias. According to the comments from the other reviewer, we also added the comparison between the  $AOD_f$  and  $AOD_t$  retrieval results and ground-based observations of different surface types. The relevant contents are shown as below:

Since our FMF is obtained from the ratio of AODf and AODt retrieval results, and the retrieval accuracy of the two parameters directly determines the retrieval accuracy of FMF, we further compared the retrieved AODs at the six different surface types with those of the ground-based data from 2006 to 2013, and the statistical results are shown in Figure R1 and Table R1. It can be seen from Figure R1 that for the comparison results of AOD6, except for the barren type, the  $AOD_{f}$  at all surface types are in good agreement with the ground-based observation results, and the r is greater than 0.7. Because the data of the barren type mainly come from the *QOMS* CAS site, the  $AOD_f$  value at this site is low, and the r is not suitable for evaluating the retrieval performance. Most of the retrieval results at barren type fall within the EE, which can indicate that the retrieval results at this type have a good accuracy. For the comparison results of  $AOD_{t}$ , the retrieval results at barren type are obviously positively shifted. This is due to the low aerosol loading at the QOMS CAS site, and the inaccurate estimation of the surface reflectance can easily magnify the errors in the retrieval results. It indicates that the EOF method used to retrieve  $AOD_t$  in this study still needs further improvement. However, it is difficult to analyse the reasons for the negative bias of most FMF retrieval results from the scatter plot, so we further counted the biases of  $AOD_t$  and  $AOD_t$ . Table R1 shows that the bias of the retrieved  $AOD_f$  and  $AOD_t$  at the six different surface types. It can be seen from Table R1 that the proportion of positive bias is greater than the proportion of negative offset for most  $AOD_t$  retrieval results, while  $AOD_f$  is the opposite. For the overall result, the bias of  $AOD_f$  is -0.037, where the proportion of negative bias is 58.68%, and the bias of  $AOD_t$  is 0.063, where the proportion of positive bias is 68.29%, indicating that the  $AOD_f$  retrieval result has a negative bias, and the AODt retrieval result has a positive bias, that is, the numerator is small and the denominator is large, eventually leading to a negative bias of FMF.

---

## Author Response (AR1)

**Responses to reviewer 1**

Thanks for your helpful comments, we have revised the paper based on your comments. The following is a one-to-one response to your comments.

**Comment:** This is an interesting study looking at the fine mode fraction (FMF) of aerosol optical depth in China. The main focus is the comparison between results from a previously published algorithm of the authors applied to POLDER data (referred to in the paper as "the algorithm"), the GRASP retrieval applied to POLDER, the MODIS Dark Target land retrieval, and the AERONET spectral deconvolution algorithm (SDA) and almucantar scan size distribution (SD) retrieval algorithm. The topic is relevant to the journal and the Special Issue. After total AOD, fine vs. coarse AOD is one of the next main frontiers of interest.

The quality of language needs some improvement. I appreciate the authors' first language is not English, and they have done a good job explaining what was done in the analysis. But some copy-editing will be necessary to bring the article to publication standards as phrasing is strange in places (too many to go through as a reviewer). This might be able to be handled by the journal production office, but if the authors have access to a service or colleague who is able to give a proof-read that would be beneficial as well. Again, the authors have done a pretty good job with the writing overall.

**Answer:** We submitted the manuscript to American Journal Experts (AJE) for editing service before submission, but we made some changes to the manuscript after the editing service was completed. This may be the cause of difficulty in reading some sentences. We have submitted the revised manuscript to AJE for editing again. Thanks for your understanding.

[Figure]

Figure R1. The editing certificate of AJE

My recommendation is for major revisions, and I would like to review the revised version.

**Major comments:**

1. It is not just the FMF which is of interest, but the overall fine and coarse AOD. It seems like a missed opportunity not to evaluate e.g. the fine mode and total AOD as well. Looking at only FMF we don't know if a bias in that is because the retrievals have errors in the total AOD or just the ratio between modes. This is briefly shown (Table 3) but only via summary metrics (would be good to see the data points) and only for the authors' approach (not MODIS or GRASP). I recommend the authors add this in the revised version. These could be also split by, for example, aerosol type or surface type (as these are some factors which often affect retrieval performance).

**Answer:** Thanks for your suggestion. We re-analyzed the corresponding AODf and AODt retrieval results from 2006 to 2013, instead of the previous result in 2013, and added the following content:

*Since our FMF is obtained from the ratio of $AOD_f$ and $AOD_t$ retrieval results, and the retrieval accuracy of the two parameters directly determines the retrieval accuracy of FMF, we further compared the retrieved AODs at the six different surface types with those of the ground-based data from 2006 to 2013, and the statistical results are shown in Figure R2 and Table R1. It can be seen from Figure R2 that for the comparison results of $AOD_f$, except for the barren type, the $AOD_f$ at all surface types are in good agreement with the ground-based observation results, and the r is greater than 0.7. Because the data of the barren type mainly come from the QOMS_CAS site, the $AOD_f$ value at this site is low, and the r is not suitable for evaluating the retrieval performance. Most of the retrieval results at barren type fall within the EE, which can indicate that the retrieval results at this type have a good accuracy. For the comparison results of $AOD_t$, the retrieval results at barren type are obviously positively shifted. This is due to the low aerosol loading at the QOMS_CAS site, and the inaccurate estimation of the surface reflectance can easily magnify the errors in the retrieval results. It indicates that the EOF method used to retrieve $AOD_t$ in this study still needs further improvement. However, it is difficult to analyse the reasons for the negative bias of most FMF retrieval results from the scatter plot, so we further counted the biases of $AOD_t$ and $AOD_f$. Table R1 shows that the bias of the retrieved $AOD_f$ and $AOD_t$ at the six different surface types. It can be seen from Table R1 that the proportion of positive bias is greater than the proportion of negative offset for most $AOD_t$ retrieval results, while $AOD_f$ is the opposite. For the overall result, the bias of $AOD_f$ is -0.037, where the proportion of negative bias is 58.68%, and the bias of $AOD_t$ is 0.063, where the proportion of positive bias is 68.29%, indicating that the $AOD_f$ retrieval result has a negative bias, and the $AOD_t$ retrieval result has a positive bias, that is, the numerator is small and the denominator is large, eventually leading to a negative bias of FMF.*

[Figure]

[Figure]

Figure R2. AODs results comparison of 6 surface types. (a), (c), (e), (g), (i), and (k) are the AOD$_t$ validation results for the type of barren, croplands, forests, grasslands, urban, and wetlands, respectively. (b), (d), (f), (h), (j), and (l) are the AOD$_f$ validation results for the type of barren, croplands, forests, grasslands, urban, and wetlands, respectively.

Table R1. Statistical analysis of $AOD_f$ and $AOD_t$ bias

| Land cover type | Retrieval parameter (550 nm) | N | r | Bias | Proportion of negative bias | Proportion of positive bias |
|---|---|---|---|---|---|---|
| Barren | $AOD_f$ | | 0.574 | 0.006 | 44.44% | 55.56% |
| | $AOD_t$ | 63 | 0.448 | 0.111 | 1.59% | 98.41% |
| | FMF | | 0.711 | -0.144 | 87.30% | 12.70% |
| Croplands | $AOD_f$ | | 0.931 | -0.038 | 55.84% | 44.16% |
| | $AOD_t$ | 394 | 0.949 | 0.077 | 27.16% | 72.84% |
| | FMF | | 0.651 | -0.064 | 64.47% | 35.53% |
| Forests | $AOD_f$ | | 0.739 | -0.049 | 64.44% | 35.56% |
| | $AOD_t$ | 45 | 0.768 | -0.019 | 48.89% | 51.11% |
| | FMF | | 0.831 | -0.102 | 75.56% | 24.44% |
| Grasslands | $AOD_f$ | | 0.892 | 0.007 | 38.05% | 61.95% |
| | $AOD_t$ | 113 | 0.841 | 0.061 | 23.89% | 76.11% |
| | FMF | | 0.777 | -0.033 | 55.75% | 44.25% |
| Urban | $AOD_f$ | | 0.906 | -0.043 | 64.61% | 35.39% |
| | $AOD_t$ | 421 | 0.926 | 0.057 | 38.72% | 61.28% |
| | FMF | | 0.733 | -0.079 | 72.45% | 27.55% |
| Wetlands | $AOD_f$ | | 0.892 | -0.065 | 69.33% | 30.67% |
| | $AOD_t$ | 150 | 0.917 | 0.048 | 37.33% | 62.67% |
| | FMF | | 0.508 | -0.031 | 55.33% | 44.67% |
| Overall | $AOD_f$ | | 0.868 | -0.037 | 58.68% | 41.32% |
| | $AOD_t$ | 1186 | 0.867 | 0.063 | 31.71% | 68.29% |
| | FMF | | 0.770 | -0.068 | 66.95% | 33.05% |

2. The authors' interpretation of MODIS Dark Target land FMF is, to my knowledge, not correct. The MODIS land fine weighting parameter eta is not a "fine mode fraction" but a "fine model fraction". The Dark Target land retrieval mixes between two bimodal size distributions, one which is mostly fine mode and one which is mostly coarse mode. See Levy 2007 (https://agupubs.onlinelibrary.wiley.com/doi/full/10.1029/2006JD007815), especially Figure 4 there. So in MODIS, FMF=0 does not mean no fine mode aerosol. It means that the proportion of the bimodal fine-dominated optical model is zero. There is still some fine mode aerosol from the coarse-dominated optical model. Similarly FMF=1 in MODIS still has some coarse aerosol present. This misinterpretation affects all the discussion of MODIS results.

**Answer:** Thanks for your correction, we do have a wrong understanding of the FMF definition of MODIS. We have rewritten the discussion about the two, and the revised part is shown as below:

*MODIS aerosol products also include FMF data sets, but this FMF has a different definition. In fact, the FMF of MODIS refers to the 'fine model fraction', which is the proportion of bimodal fine-dominated aerosol model, but not pure fine mode (Levy et al., 2007). Because the FMF results obtained by MODIS are different in definition from the ground-based results (Levy et al., 2009), the retrieval results are quite different from the ground-based observation results, which limits the research that depends on the FMF parameter. We compared the retrieved and*

*MODIS FMF with the AERONET ground-based observations to further evaluate the significance of our results. The MODIS FMF results were derived from the MYD04 product of collection 6.1. Figure 7 shows the comparison between the two results and the AERONET ground-based observation results from 2011 to 2013, which is the results where both MODIS and POLDER matching the ground-based observations. As seen from the figure, compared with ground-based observations, the r of FMF obtained in this study is 0.812, while that of MODIS is 0.302. The correlation coefficient of the results obtained in this study is much higher than that of MODIS. At the same time, notice that there are many 0 values in the MODIS results. These 0 values are not meaningless but correspond to the situation where there is no the fine-dominated aerosol model.*

*More statistical results of the two are shown in Table 4. The table shows that the FMF results obtained in this study have an MAE of 0.072, an RMSE of 0.102, and a Within EE of 87.41%, whereas results of MODIS have an MAE of 0.512, RMSE of 0.574, and Within EE of 19.58%. The statistical indicators of the FMF results obtained by our study are closer to the ground-based observations than the MODIS results. Nevertheless, note that this does not mean that the FMF of MODIS has a large deviation. As mentioned above, there is a difference in definition between the FMF of MODIS and the ground-based observations; consequently, it is difficult to obtain the true deviation of MODIS FMF based on ground-based observations.*

References:

Levy, R. C., Remer, L. A., and Dubovik, O.: Global aerosol optical properties and application to Moderate Resolution Imaging Spectroradiometer aerosol retrieval over land, Journal of Geophysical Research: Atmospheres, 112, https://doi.org/10.1029/2006JD007815, 2007.

Levy, R. C., Remer, L. A., Tanré, D., Mattoo, S., Vermote, E. F., and Kaufman, Y. J.: Algorithm for Remote Sensing of Tropospheric Aerosol over Dark Targets from MODIS:Collections 005 and 051: Revision 2, 2009.

3. The definition of FMF in other products is not be the same either, e.g. the AERONET SDA assumes a combination of fine and coarse modes but the AERONET sky-scan retrieval and (I think) GRASP look for a minimum in the size distribution and make the fine/coarse split there. The manuscript should be more detailed about the exact definition of fine mode fraction within the products, and make sure they are comparable. If not then the discrepancies will be partly due to definition differences rather than retrieval problems.

**Answer:** We agree that different products in this study have different FMF definitions, We have added the following discussion about FMF definitions in Section 3.3:

*GRASP products provide $AOD_f$ and $AOD_t$ datasets, but do not directly provide FMF datasets. In this study, the ratio of the two was used to obtain the GRASP FMF. However, it should be noted that the definition of GRASP $AOD_f$ is somewhat different from the $AOD_f$ in our research, which may eventually lead to the difference in the definition of FMF. The $AOD_f$ in our study is similar to the definition in the ground-based SDA algorithm; there is no clear cut-off particle size, that is, its definition is indefinite. This is different from the $AOD_f$ obtained by calculating and integrating the size distribution in GRASP, so the difference in the spatial distribution results of the two may be caused by the definition, rather than a problem in the retrieval algorithm.*

**Minor comments:**

1. Line 98: is the EOF method similar to the MISR land approach? That could be mentioned (and compared if different) as it is likely that the readership of this journal would have some familiarity with it.

**Answer:** Yes. The EOF method used in our retrieval is similar to the MISR approach, we transplanted this method to POLDER. We have added the following relevant information in the revised paper:

*The EOF method has previously been used for the retrieval of land aerosols on Multi-angle Imaging Spectro Radiometer (MISR); we transplanted this method to POLDER based on the MISR approach. For more details, please refer to our 2017 study (Zhang et al., 2017)*

References:

Zhang, Y., Li, Z., Qie, L., Hou, W., Liu, Z., Zhang, Y., Xie, Y., Chen, X., and Xu, H.: Retrieval of Aerosol Optical Depth Using the Empirical Orthogonal Functions (EOFs) Based on PARASOL Multi-Angle Intensity Data, Remote Sensing, 2017, 578, 2017.

2. Line 133: Angstrom should be written Ångström here.
**Answer:** We have modified 'Angstrom' to 'Ångström'.

3. Line 154: I don't know why it makes sense for EE to be +/-(0.1+10%). Why should FMF uncertainty depend on FMF? Is high FMF slightly harder to retrieve? More justification is needed here. If this was used in a previous study, we need to see the justification there, and if there wasn't one, then that's an issue. I do not see a physical reasoning why FMF uncertainty should be a function of FMF.

**Answer:** We have not considered this issue carefully before, but habitually apply the way of EE definition of AOD to FMF. However, there does not seem to be a unified standard for EE definition of FMF, different studies have different standards. For example, the study of Cheng et al. did not define the EE of FMF. The study of Yan et al. defined the EE of FMF as ±0.4. The study of Chen et al. defines three types of FMF EE: +/-(0 +40%), +/-(0 +25%), +/-(0.03 +20%). After carefully considering your comments, we think that the EE of FMF should not be a function of FMF, so we changed the EE of FMF in this study to +/-0.2, which considered our original absolute error (0.1) in the EE and the uncertainties of AERONET FMF (be of order 0.1 to 0.15, we used a value of 0.1) mentioned in your next comment. The 'EE lines' and 'Within EE' on all the scatter plots were modified.

References:

T, Cheng, X, et al. Aerosol optical depth and fine-mode fraction retrieval over East Asia using multi-angular total and polarized remote sensing[J]. Atmospheric Measurement Techniques, 2012.

Yan X, Li Z , Shi W , et al. An improved algorithm for retrieving the fine-mode fraction of aerosol optical thickness, part 1: Algorithm development[J]. Remote Sensing of Environment, 2017, 192:87-97.

Chen, X., de Leeuw, G., Arola, A., Liu, S., Liu, Y., Li, Z., and Zhang, K.: Joint retrieval of the

aerosol fine mode fraction and optical depth using MODIS spectral reflectance over northern and eastern China: Artificial neural network method, Remote Sensing of Environment, 249, 112006, 2020

O'Neill, Norm T, Dubovik, et al. Modified Ångström Exponent for the Characterization of Submicrometer Aerosols[J]. Applied Optics, 2001.

4. Line 154: Also, when calculating this metric, the uncertainty on AERONET FMF should be accounted for as well (this is dependent on AOD but can also be of order 0.1 to 0.15: this is discussed in some AERONET publications). Unlike AERONET total AOD, AERONET FMF cannot be considered a reference truth because there are non-negligible uncertainties in both the AERONET SDA and SD retrievals.

**Answer:** As in the previous answer, we changed the EE of FMF to +/-0.2.

5. Lines 220-221: I believe the latest GRASP is version 2.1, not 2.0.6 as stated here. Also, which GRASP product? There are 3 separate GRASP POLDER data sets with different assumptions about aerosol size distribution form. See this paper by Chen et al for more information: https://essd.copernicus.org/preprints/essd-2020-224/. More information should be added to the manuscript, and if possible the analysis should use the latest GRASP version. Including those results in this paper (rather than just citing an older evaluation study) would also help to compare GRASP and the authors' new approach. Right now it is still not clear to me which is better or what the relative benefits are.

**Answer:** Yes, the latest GRASP is version 2.1 according to the paper by Chen et al., and the GRASP product used in our previous study is from the «high-precision» approach. However, the latest version can be obtained from AERIS/ICARE Data and Services Center (http://www.icare.univ-lille.fr) is version 2.06 (Figure R3), we can only use this older version of the product for processing. We have added the following relevant information in the revised paper:

*In our previous research, the accuracy of FMF calculated from the GRASP «high-precision» product was validated.*

*The GRASP product version we processed is V2.06, which is the latest version that can be obtained from AERIS/ICARE Data and Services Center (http://www.icare.univ-lille.fr; last accessed on December 27, 2020).*

Chen et al. did not discuss the uncertainty of FMF of GRASP, and we cannot directly compare the two FMF based on their research. But we have added a discussion about the FMF definition as below:

*GRASP products provide $AOD_f$ and $AOD_t$ datasets, but do not directly provide FMF datasets. In this study, the ratio of the two was used to obtain the GRASP FMF. However, it should be noted that the definition of GRASP $AOD_f$ is somewhat different from the $AOD_f$ in our research, which may eventually lead to the difference in the definition of FMF. The $AOD_f$ in our study is similar to the definition in the ground-based SDA algorithm; there is no clear cut-off particle size, that is, its definition is indefinite. This is different from the $AOD_f$ obtained by calculating and integrating the size distribution in GRASP, so the difference in the spatial distribution results of the two may be caused by the definition, rather than a problem in the retrieval algorithm. In the research of Chen et al. (Chen et al., 2020), in their comparison with*

*AERONET observations, the r of $AOD_f$ is between 0.868 (models approach) and 0.924 (high-precision approach), which is similar to the r (0.868) of $AOD_f$ in this study, but their bias is only -0.02 (models approach) and 0.01 (high-precision approach), which is different from the bias (-0.037) of $AOD_f$ in this study. This indicates that the definition of $AOD_f$ in GRASP and our study may be different.*

Figure R3. The list of datasets on ICARE data center

Reference:

Chen, C., Dubovik, O., Fuertes, D., Litvinov, P., Lapyonok, T., Lopatin, A., Ducos, F., Derimian, Y., Herman, M., Tanré, D., Remer, L. A., Lyapustin, A., Sayer, A. M., Levy, R. C., Hsu, N. C., Descloitres, J., Li, L., Torres, B., Karol, Y., Herrera, M., Herreras, M., Aspetsberger, M., Wanzenboeck, M., Bindreiter, L., Marth, D., Hangler, A., and Federspiel, C.: Validation of GRASP algorithm product from POLDER/PARASOL data and assessment of multi-angular polarimetry potential for aerosol monitoring, Earth Syst. Sci. Data Discuss., 2020, 1-108, 10.5194/essd-2020-224, 2020.

6. Line 243: if possible, more information about the PM2.5 and PM10 surface measurements should be made here. For example is this BAM, filter, or something else?

**Answer:** Sorry, the National Bureau of Statistics has not described the PM2.5 and PM10 measurement method in the Statistical Yearbook. As far as I know, the environmental monitoring national control station in China uses two methods, the beta ray method and the oscillatory balance method, but it is uncertain which method is used for the specific station.

7. Section 4.1: I do not see much value in showing these two case studies. It's just a couple of maps and text describing them. There isn't really enough context or external data sets brought in to make them interesting. I recommend removing section 4.1 (so section 4.2 would just be called section 4), unless the authors can provide additional material of scientific interest to make the reader care about these examples.

**Answer:** We accepted your comment and deleted this section.

8. Section 4.2: It would be good to add AOD maps here as well, for additional context

**Answer:** We accepted your suggestion and finally deleted the original section 4.

9. Figure 1: "Fuction" should say "Function".

**Answer:** We have corrected it.

10. Figure 2: I don't know why there is a color bar (FMF) on this figure, given it is only showing site locations on a blank background map

**Answer:** We have corrected it.

11. Figure 3: These panels should have separate titles or similar (e.g. a, b) to separate them. Also, I would remove the regression lines. I don't think they add anything, and don't think they are appropriate. In some cases the relationships aren't linear (e.g. QOMS_CAS), and the technique is not valid because (1) it is not accounting for uncertainty in the AERONET reference data and (2) the data are constrained by the possible bounds of FMF (i.e. 0-1) meaning that errors cannot be Gaussian and unbiased. Both of these means that the assumptions required for validation are not satisfied.

**Answer:** The separate titles are in the lower right corner of the figures. We accepted your suggestion and deleted the regression lines. We originally wanted to use a regression line to represent the deviation between the retrieval result and the AERONET result. We did not consider the error of AERONET FMF itself, and did not consider the normal distribution issue, but used it as a true value. The revised figures (Figure R4) are shown as below:

[Figure]

[Figure]

[Figure]

Figure R4. FMF results comparison at 14 AERONET sites. (a) - (n) are the validation results for the Beijing, Hangzhou_city, Hongkong_PolyU, Kaiping, Lanzhou_city, NAM_CO, NUIST, QOMS_CAS, SACOL, Taihu, Taipei, Xianghe, Xinglong, Zhongshan_Univ sites, respectively.

The corresponding 'within EE' in Table 2 has also been modified:

Table R2. FMF validation results of different surface types

| Land cover type | N | r | MAE | RMSE | Within EE |
|---|---|---|---|---|---|
| Overall result | 1186 | 0.770 | 0.143 | 0.170 | 65.01% |
| Urban | 421 | 0.733 | 0.139 | 0.163 | 66.98% |
| Barren | 63 | 0.711 | 0.158 | 0.182 | 55.55% |
| Grasslands | 113 | 0.777 | 0.137 | 0.170 | 66.37% |
| Wetlands | 150 | 0.508 | 0.145 | 0.176 | 63.33% |
| Croplands | 394 | 0.651 | 0.146 | 0.174 | 64.21% |
| Forests | 45 | 0.831 | 0.133 | 0.159 | 68.88% |

12. Figure 4: It would be good to show a second histogram in addition, filtered for points where the AOD is above a certain value (e.g. 0.2?). We would expect that this would be thinner because the sensitivity to FMF should be better when AOD is high. So this would be interesting to see how the width and midpoint of the distribution change.

**Answer:** In accordance with your opinion, we have added the FMF error distribution result when $AOD_f$ is greater than 0.2 (Figure R5). Comparing the two results, it can be found that after screening, the proportion of FMF error ranging from -0.4 to -0.3 decreased by about 7), and the proportion of FMF error ranging from -0.1 to 0.1 increased by about 6%, which shows that when the AOD is higher, our FMF retrieval method is more sensitive.

[Figure]

Figure R5. FMF retrieval error distribution results. (a) is for all results, (b) is for the results with AODf greater than 0.2.

13. Figure 5: same comment about regression line as for Figure 3. Also, both data sets have 143 points here: is this for the points where MODIS and POLDER are all matched together? This should be stated in the paper, this was not clear to me.

**Answer:** We have deleted the regression line, and the revised figure (Figure R6) is shown as below. This is for the points where MODIS and POLDER are all matched together. We have stated it in the revised paper as below:

*Figure R6 shows the comparison between the two results and the AERONET ground-based observation results from 2011 to 2013, which is the results where both MODIS and POLDER matching the ground-based observations.*

[Figure]

Figure R6. Comparison between the results of this study and MODIS FMF with AERONET

14. Figures 6-8: can we have more than 3 points labeled on the color scale? Also, it would be clearer to combine these together into one 1 figure, possibly with figure 9 as well.

**Answer:** Now there are 6 points labeled on the color scale, and we combined the original

figures 6-9 into one figure (Figure R7) as shown below:

[Figure]

Figure R7. Distribution of FMF of China in 2013 from different data sources. (a) is the normalized results of this study, (b) is the normalized results of MODIS, (c) is the normalized results of GRASP, and (d) is the GRASP results minus the retrieved results (non-normalized).

15. Figure 9: is this FMF difference or normalized FMF difference? This was not clear to me.
**Answer:** This is non-normalized FMF difference. We have modified the figure and the corresponding description in the paper as shown in the previous answer.

16. Figure 10: same comment about regression line as for Figure 3.
**Answer:** We have deleted the regression line, and the revised figure (Figure R8) is shown as below:

[Figure]

Figure R8. Comparison between the ratio of PM2.5 to PM10 and FMF (hourly average)

17. Figures 15, 16: I don't see much value in these figures and suggest deleting them. The overall spatial distribution looks fairly similar year to year, and I think the seasonal maps in Figure 17 are more useful. Likewise, I am not convinced that the differences in Figure 16 are realistic. It looks like a FMF difference of +0.2 across many parts of China, even remote areas where the aerosol is mostly dust. So in my view it could easily be a calibration drift, as POLDER had no on-board calibration. There is only a paragraph devoted to Figure 16 anyway. If the authors wish to discuss trends, it would make more sense to show also total AOD and fine mode AOD (so we can see which is increasing) and bring in some additional satellite, model, or ground-based data to help verify and understand the mechanisms. If Figures 15 and 16 are removed, then Figure 17 could also be moved earlier in the manuscript, close to where the authors' retrieval method is introduced.

**Answer:** We accepted your suggestion and considered that the analysis on AOD and FMF may be written as a separate paper, so Figure 17 and related content are moved after the method introduction. In this way, the revised paper does not have the original section 4, the focus of the whole paper is the validation and comparison of the FMF results, and the corresponding paper title is also revised as 'Retrieval of Aerosol Fine-mode Fraction over China from Satellite Multiangle Polarized Observations: Validation and Comparison'.

**Responses to reviewer 2**

Thanks for your helpful comments, we have revised the paper based on your comments. The following is a one-to-one response to your comments.

GENERAL COMMENTS

This manuscript by Zhang et al. conducted the fine mode fraction (FMF) retrieval from muliti-angular polarimeter (PARASOL). Technically, the total AOD is determined from intensity measurements, and fine mode AOD is derived from multi-angular polarized measurements. Then the ratio of AOD and fine mode AOD derives FMF. This method generally sounds, and has been published in Zhang et al. (2017, 2018). This manuscript is mainly focus on the validation of retrieved FMF using AERONET, MODIS, PARASOL/GRASP products. The main concern here is that each product may have different definition of their FMF, this should be fully considered before conducting validation and inter-comparison. For example, MODIS FMF over land is the ratio to reflectance instead of total AOD; therefore MODIS FMF over land has little physical meaning. Over ocean, by single scattering approximation, FMF can be approximated as weighted for AOD (see discussions in Remer et al., 2005). Additionally, the objective is not clear why the authors pay close attention to FMF instead of fine mode AOD, the uncertainties in both AOD and fine AOD could significantly worsen the FMF quality, and a good FMF doesn't necessarily produce a good estimation of fine mode AOD. Overall, I think this manuscript is within the scope of AMT. Some comments and concerns are required to be addressed and clearly stated before being published. The specific comments are listed as follow.

In the revised paper, we have discussed the differences in the definition of different FMF products. Please check our revised paper later. This part is also included in our answer to your comment below. In 2015, we proposed the PMRS model (Zhang et al., 2015), which is a model based on physical methods to estimate PM2.5 concentration. In that model, FMF is an important input parameter and cannot be replaced by AOD$_f$. Since the existing MODIS FMF products are difficult to meet the application requirements of the PMRS model, we started the research of using multi-angle polarization sensors to retrieve FMF. In addition, FMF can also be used to distinguish anthropogenic and natural aerosol types (Bellouin et al., 2005). We think that FMF is also important for research in the field of atmospheric environment.

Line 158: Section name is wrong.

**Answer:** We have modified the section name as 'Validation against AERONET ground-based data'.

Figure 3: is this all points from 2006-2013? Any filter scheme used, please clarify.

**Answer:** Yes, this is all the matched points from 2006 to 2013. When the retrieved $AOD_f$ is greater than the retrieved $AOD_t$, we consider this situation as a failure of the FMF retrieval, and the results of this part were not involved in the comparison. These results account for about 10% We have added those information in section 2.3 as follows:

*Note that when the retrieved $AOD_f$ is greater than the retrieved $AOD_t$, we consider this situation as a failure of the FMF retrieval, and the results of this part were not involved in the comparison. These results account for about 10%.*

Line 177: errors : : : are stable: : : ?? please consider 'uncertainty'.

**Answer:** We have corrected it.

Line 182: the definitions of AERONET FMF and retrieved AODf/AOD are not identical.

**Answer:** We agree that the definitions of the AERONET FMF and retrieved FMF are not identical. However, they have some similarities. The definition of the $AOD_f$ in our study is indefinite, and it has no clear cut-off particle size. Similarly, there is also no clear definition of $AOD_f$ in the ground-based SDA algorithm. Therefore, we think that although they are not equivalent, the two are comparable. We prefer to use the SDA FMF as the 'truth value' for validation.

Table 3: Number of points is critical, as well as other parameters (r, rmse, etc.).

**Answer:** We have added the information of the number of points, r and bias. According to the comments from the other reviewer, we also added the comparison between the $AOD_f$ and $AOD_t$ retrieval results and ground-based observations of different surface types. The relevant contents are shown as below:

*Since our FMF is obtained from the ratio of $AOD_f$ and $AOD_t$ retrieval results, and the retrieval accuracy of the two parameters directly determines the retrieval accuracy of FMF, we further compared the retrieved AODs at the six different surface types with those of the ground-based data from 2006 to 2013, and the statistical results are shown in Figure R1 and Table R1. It can be seen from Figure R1 that for the comparison results of $AOD_f$, except for the barren type, the $AOD_f$ at all surface types are in good agreement with the ground-based observation results, and the r is greater than 0.7. Because the data of the barren type mainly come from the QOMS_CAS site, the $AOD_f$ value at this site is low, and the r is not suitable for evaluating the retrieval performance. Most of the retrieval results at barren type fall within the EE, which can indicate that the retrieval results at this type have a good accuracy. For the comparison results of $AOD_t$, the retrieval results at barren type are obviously positively shifted. This is due to the low aerosol loading at the QOMS_CAS site, and the inaccurate estimation of the surface reflectance can easily magnify the errors in the retrieval results. It indicates that the EOF method used to retrieve $AOD_t$ in this study still needs further improvement. However, it is difficult to analyse the reasons for the negative bias of most FMF retrieval results from the scatter plot, so we further counted the biases of $AOD_t$ and $AOD_f$. Table R1 shows that the bias of the retrieved $AOD_f$ and $AOD_t$ at the six different surface types. It can be seen from Table R1 that the proportion of positive bias is greater than the proportion of negative offset for most $AOD_t$ retrieval results, while $AOD_f$ is the opposite. For the overall result, the bias of $AOD_f$ is -0.037, where the proportion of negative bias is 58.68%, and the bias of $AOD_t$ is 0.063, where the proportion of positive bias is 68.29%, indicating that the $AOD_f$ retrieval result has a negative bias, and the $AOD_t$ retrieval result has a positive bias, that is, the numerator is small and the denominator is large, eventually leading to a negative bias of FMF.*

[Figure]

[Figure]

Figure R1. AODs results comparison of 6 surface types. (a), (c), (e), (g), (i), and (k) are the AOD$_t$ validation results for the type of barren, croplands, forests, grasslands, urban, and wetlands, respectively. (b), (d), (f), (h), (j), and (l) are the AOD$_f$ validation results for the type of barren, croplands, forests, grasslands, urban, and wetlands, respectively.

The final revised table is shown as below:

Table R1. Statistical analysis of AOD$_f$ and AOD$_t$ bias

| Land cover | Retrieval | N | r | Bias | Proportion of | Proportion of |
|---|---|---|---|---|---|---|

| type | parameter (550 nm) | | | | negative bias | positive bias |
|---|---|---|---|---|---|---|
| Barren | AOD$_f$ | | 0.574 | 0.006 | 44.44% | 55.56% |
| | AOD$_t$ | 63 | 0.448 | 0.111 | 1.59% | 98.41% |
| | FMF | | 0.711 | -0.144 | 87.30% | 12.70% |
| Croplands | AOD$_f$ | | 0.931 | -0.038 | 55.84% | 44.16% |
| | AOD$_t$ | 394 | 0.949 | 0.077 | 27.16% | 72.84% |
| | FMF | | 0.651 | -0.064 | 64.47% | 35.53% |
| Forests | AOD$_f$ | | 0.739 | -0.049 | 64.44% | 35.56% |
| | AOD$_t$ | 45 | 0.768 | -0.019 | 48.89% | 51.11% |
| | FMF | | 0.831 | -0.102 | 75.56% | 24.44% |
| Grasslands | AOD$_f$ | | 0.892 | 0.007 | 38.05% | 61.95% |
| | AOD$_t$ | 113 | 0.841 | 0.061 | 23.89% | 76.11% |
| | FMF | | 0.777 | -0.033 | 55.75% | 44.25% |
| Urban | AOD$_f$ | | 0.906 | -0.043 | 64.61% | 35.39% |
| | AOD$_t$ | 421 | 0.926 | 0.057 | 38.72% | 61.28% |
| | FMF | | 0.733 | -0.079 | 72.45% | 27.55% |
| Wetlands | AOD$_f$ | | 0.892 | -0.065 | 69.33% | 30.67% |
| | AOD$_t$ | 150 | 0.917 | 0.048 | 37.33% | 62.67% |
| | FMF | | 0.508 | -0.031 | 55.33% | 44.67% |
| Overall | AOD$_f$ | | 0.868 | -0.037 | 58.68% | 41.32% |
| | AOD$_t$ | 1186 | 0.867 | 0.063 | 31.71% | 68.29% |
| | FMF | | 0.770 | -0.068 | 66.95% | 33.05% |

Line 220: Please identify products name and version, and last access, etc. (This is necessary for all products used in the manuscript)

**Answer:** We have added the GRASP products information as follows:

*The GRASP product version we processed is V2.06, which is the latest version that can be obtained from AERIS/ICARE Data and Services Center ([http://www.icare.univ-lille.fr](http://www.icare.univ-lille.fr); last accessed on December 27, 2020).*

Line 222: what do you mean normalized FMF?

**Answer:** To facilitate the comparison of the differences in the spatial distribution trends of those results from this study, MODIS and GRASP, all the results are normalized, meaning they are divided by the maximum value in the respective FMF image.

Section 3.3: why only 2013 data is compared? It would be interesting to check more data 2006-2013 and other related parameters, e.g. AOD and fine mode AOD, to make the conclusion more solid.

**Answer:** Due to the limited ground PM2.5/PM10 data, we can only compare the results in 2013. As shown in Figure R1 and Table R1, we compared the retrieved AOD$_f$ and AOD$_t$ with those from the ground-based observations. We also added the following discussion about FMF definitions of this study and GRASP in Section 3.3:

*GRASP products provide AOD$_f$ and AOD$_t$ datasets, but do not directly provide FMF datasets. In this study, the ratio of the two was used to obtain the GRASP FMF. However, it should be*

*noted that the definition of GRASP AOD$_f$ is somewhat different from the AOD$_f$ in our research, which may eventually lead to the difference in the definition of FMF. The AOD$_f$ in our study is similar to the definition in the ground-based SDA algorithm; there is no clear cut-off particle size, that is, its definition is indefinite. This is different from the AOD$_f$ obtained by calculating and integrating the size distribution in GRASP, so the difference in the spatial distribution results of the two may be caused by the definition, rather than a problem in the retrieval algorithm. In the research of Chen et al. (Chen et al., 2020), in their comparison with AERONET observations, the r of AOD$_f$ is between 0.868 (models approach) and 0.924 (high-precision approach), which is similar to the r (0.868) of AOD$_f$ in this study, but their bias is only -0.02 (models approach) and 0.01 (high-precision approach), which is different from the bias (-0.037) of AOD$_f$ in this study. This indicates that the definition of AOD$_f$ in GRASP and our study may be different.*

Reference:

Chen, C., Dubovik, O., Fuertes, D., Litvinov, P., Lapyonok, T., Lopatin, A., Ducos, F., Derimian, Y., Herman, M., Tanré, D., Remer, L. A., Lyapustin, A., Sayer, A. M., Levy, R. C., Hsu, N. C., Descloitres, J., Li, L., Torres, B., Karol, Y., Herrera, M., Herreras, M., Aspetsberger, M., Wanzenboeck, M., Bindreiter, L., Marth, D., Hangler, A., and Federspiel, C.: Validation of GRASP algorithm product from POLDER/PARASOL data and assessment of multi-angular polarimetry potential for aerosol monitoring, Earth Syst. Sci. Data Discuss., 2020, 1-108, 10.5194/essd-2020-224, 2020.

Figures 6, 7, 8: it is important to mention the spatial resolution, visually, the derived FMF in figure 6 has much coarser resolution than others.

**Answer:** We have added the spatial resolution information of the corresponding result in the figure title. According to the suggestion from the other reviewer, we integrated the original Figure 6-9 into one Figure (Figure R2).

[Figure]

Figure R2. Distribution of FMF of China in 2013 from different sources. (a) is the normalized results of this study (18 km resolution), (b) is the normalized results of MODIS (10 km resolution), (c) is the normalized results of GRASP (6 km resolution), and (d) is the GRASP results minus the retrieved results (non-normalized, 18 km resolution).

Figures9, 16: the quality of figures showing differences can be improved by using more adequate colorbar.

**Answer:** There are 5-6 points labeled on the color scale now (Figure R2). However, we only retained the seasonal average spatial distribution results of FMF in the revised paper according to the comments from the other reviewer, and the original Figure 16 has been deleted.

Line 346: throughout the manuscript, no place specified the MODIS (TERRA or AQUA or both) dataset.

**Answer:** We have added the information of the MODIS FMF results in section 3.2 as follows:
*The MODIS FMF results were derived from the MYD04 product of collection 6.1.*

Line 370: Is there any specific reason to pay close attention to FMF instead of fine mode AOD? On one hand, the uncertainties in both AOD and fine AOD could significantly worsen the FMF, on the other hand, a good FMF doesn't necessarily produce a good estimation of fine mode AOD, which can compensate by AOD and fine AOD, right?

**Answer:** In 2015, we proposed the PMRS model (Zhang et al., 2015), which is a model based on physical methods to estimate $PM_{2.5}$ concentration. In that model, FMF is an important input parameter and cannot be replaced by $AOD_f$. Since the existing MODIS FMF products are difficult to meet the application requirements of the PMRS model, we started the research of using multi-angle polarization sensors to retrieve FMF. In addition, FMF can also be used to distinguish

anthropogenic and natural aerosol types (Bellouin et al., 2005). We think that FMF is also important for research in the field of atmospheric environment. We have rewritten that sentence as follows:

[revised manuscript text omitted]